# Error Bounds for Flow Matching Methods

**Joe Benton**  *benton@stats.ox.ac.uk*
*Department of Statistics*
*University of Oxford*

**George Deligiannidis**  *deligian@stats.ox.ac.uk*
*Department of Statistics*
*University of Oxford*

**Arnaud Doucet**  *doucet@stats.ox.ac.uk*
*Department of Statistics*
*University of Oxford*

**Reviewed on OpenReview:** *https://openreview.net/forum?id=uqQPyWFDhY*

## Abstract

Score-based generative models are a popular class of generative modelling techniques relying on stochastic differential equations (SDEs). From their inception, it was realized that it was also possible to perform generation using ordinary differential equations (ODEs) rather than SDEs. This led to the introduction of the probability flow ODE approach and denoising diffusion implicit models. Flow matching methods have recently further extended these ODE-based approaches and approximate a flow between two arbitrary probability distributions. Previous work derived bounds on the approximation error of diffusion models under the stochastic sampling regime, given assumptions on the $L^2$ loss. We present error bounds for the flow matching procedure using fully deterministic sampling, assuming an $L^2$ bound on the approximation error and a certain regularity condition on the data distributions.

## 1 Introduction

Much recent progress in generative modelling has focused on learning a map from an easy-to-sample reference distribution $\pi_0$ to a target distribution $\pi_1$. Recent works have demonstrated how to learn such a map by defining a stochastic process transforming $\pi_0$ into $\pi_1$ and learning to approximate its marginal vector flow, a procedure known as flow matching (Lipman et al., 2023; Liu et al., 2023; Albergo & Vanden-Eijnden, 2023; Albergo et al., 2023; Heitz et al., 2023). Score-based generative models (Song et al., 2021b; Song & Ermon, 2019; Ho et al., 2020) can be viewed as a particular instance of flow matching where the interpolating paths are defined via Gaussian transition densities. However, the more general flow matching setting allows one to consider a broader class of interpolating paths, and leads to deterministic sampling schemes which are typically faster and require fewer steps (Song et al., 2021a; Zhang & Chen, 2023).

Given the striking empirical successes of these models at generating high-quality audio and visual data (Dhariwal & Nichol, 2021; Popov et al., 2021; Ramesh et al., 2022; Saharia et al., 2022), there has been significant interest in understanding their theoretical properties. Several works have sought convergence guarantees for score-based generative models with stochastic sampling (Block et al., 2022; De Bortoli, 2022; Lee et al., 2022), with recent work demonstrating that the approximation error of these methods decays polynomially in the relevant problem parameters under mild assumptions (Chen et al., 2023c; Lee et al., 2023; Chen et al., 2023a). Other works have explored bounds in the more general flow matching setting (Albergo & Vanden-Eijnden, 2023; Albergo et al., 2023). However these works either still require some stochasticity in the sampling procedure or do not hold for data distributions without full support.

In this work we present the first bounds on the error of the flow matching procedure that apply with fully deterministic sampling for data distributions without full support. Our results come in two parts: first, we control the error of the flow matching approximation under the 2-Wasserstein metric in terms of the $L^2$ training error and the Lipschitz constant of the approximate velocity field; second, we show that, under a smoothness assumption explained in Section 3.2, the true velocity field is Lipschitz and we bound the associated Lipschitz constant. Combining the two results, we obtain a bound on the approximation error of flow matching which depends polynomially on the $L^2$ training error.

## 1.1 Related work

**Flow methods:** Probability flow ODEs were originally introduced by Song et al. (2021b) in the context of score-based generative modelling. They can be viewed as an instance of the normalizing flow framework (Rezende & Mohamed, 2015; Chen et al., 2018), but where the additional Gaussian diffusion structure allows for much faster training and sampling schemes, for example using denoising score matching (Vincent, 2011), compared to previous methods (Grathwohl et al., 2019; Rozen et al., 2021; Ben-Hamu et al., 2022).

Diffusion models are typically expensive to sample and several works have introduced simplified sampling or training procedures. Song et al. (2021a) propose Denoising Diffusion Implicit Models (DDIM), a method using non-Markovian noising processes which allows for faster deterministic sampling. Alternatively, (Lipman et al., 2023; Liu et al., 2023; Albergo & Vanden-Eijnden, 2023) propose flow matching as a technique for simplifying the training procedure and allowing for deterministic sampling, while also incorporating a much wider class of possible noising processes. Albergo et al. (2023) provide an in depth study of flow matching methods, including a discussion of the relative benefits of different interpolating paths and stochastic versus deterministic sampling methods.

**Error bounds:** Bounds on the approximation error of diffusion models with stochastic sampling procedures have been extensively studied. Initial results typically relied either on restrictive assumptions on the data distribution (Lee et al., 2022; Yang & Wibisono, 2022) or produced non-quantitative or exponential bounds (Pidstrigach, 2022; Liu et al., 2022; De Bortoli et al., 2021; De Bortoli, 2022; Block et al., 2022). Recently, several works have derived polynomial convergence rates for diffusion models with stochastic sampling (Chen et al., 2023c; Lee et al., 2023; Chen et al., 2023a; Li et al., 2023; Benton et al., 2023; Conforti et al., 2023).

By comparison, the deterministic sampling regime is less well explored. On the one hand, Albergo & Vanden-Eijnden (2023) give a bound on the 2-Wasserstein distance between the endpoints of two flow ODEs which depends on the Lipschitz constants of the flows. However, their Lipschitz constant is uniform in time, whereas for most practical data distributions we expect the Lipschitz constant to explode as one approaches the data distribution; for example, this will happen for data supported on a submanifold. Additionally, Chen et al. (2023d) derive a polynomial bound on the error of the discretised exact reverse probability flow ODE, though their work does not treat learned approximations to the flow. Li et al. (2023) also provide bounds for fully deterministic probability flow sampling, but also require control of the difference between the derivatives of the true and approximate scores.

On the other hand, most other works studying error bounds for flow methods require at least some stochasticity in the sampling scheme. Albergo et al. (2023) provide bounds on the Kullback–Leibler (KL) error of flow matching, but introduce a small amount of random noise into their sampling procedure in order to smooth the reverse process. Chen et al. (2023b) derive polynomial error bounds for probability flow ODE with a learned approximation to the score function. However, they must interleave predictor steps of the reverse ODE with corrector steps to smooth the reverse process, meaning that their sampling procedure is also non-deterministic.

In contrast, we provide bounds on the approximation error of a fully deterministic flow matching sampling scheme, and show how those bounds behave for typical data distributions (for example, those supported on a manifold).

## 2 Background on flow matching methods

We give a brief overview of flow matching, as introduced by Lipman et al. (2023); Liu et al. (2023); Albergo & Vanden-Eijnden (2023); Heitz et al. (2023). Given two probability distributions $\pi_0$, $\pi_1$ on $\mathbb{R}^d$, flow matching is a method for learning a deterministic coupling between $\pi_0$ and $\pi_1$. It works by finding a time-dependent vector field $v : \mathbb{R}^d \times [0, 1] \to \mathbb{R}^d$ such that if $Z^{\mathbf{x}} = (Z_t^{\mathbf{x}})_{t \in [0,1]}$ is a solution to the ODE

$$\frac{\mathrm{d}Z_t^{\mathbf{x}}}{\mathrm{d}t} = v(Z_t^{\mathbf{x}}, t), \quad Z_0^{\mathbf{x}} = \mathbf{x} \tag{1}$$

for each $\mathbf{x} \in \mathbb{R}^d$ and if we define $Z = (Z_t)_{t \in [0,1]}$ by taking $\mathbf{x} \sim \pi_0$ and setting $Z_t = Z_t^{\mathbf{x}}$ for all $t \in [0, 1]$, then $Z_1 \sim \pi_1$. When $Z$ solves (1) for a given function $v$, we say that $Z$ is a flow with velocity field $v$. If we have such a velocity field, then $(Z_0, Z_1)$ is a coupling of $(\pi_0, \pi_1)$. If we can sample efficiently from $\pi_0$ then we can generate approximate samples from the coupling by sampling $Z_0 \sim \pi_0$ and numerically integrating (1). This setup can be seen as an instance of the continuous normalizing flow framework (Chen et al., 2018).

In order to find such a vector field $v$, flow matching starts by specifying a path $I(\mathbf{x}_0, \mathbf{x}_1, t)$ between every two points $\mathbf{x}_0$ and $\mathbf{x}_1$ in $\mathbb{R}^d$. We do this via an interpolant function $I : \mathbb{R}^d \times \mathbb{R}^d \times [0, 1] \to \mathbb{R}^d$, which satisfies $I(\mathbf{x}_0, \mathbf{x}_1, 0) = \mathbf{x}_0$ and $I(\mathbf{x}_0, \mathbf{x}_1, 1) = \mathbf{x}_1$. In this work, we will restrict ourselves to the case of spatially linear interpolants, where $I(\mathbf{x}_0, \mathbf{x}_1, t) = \alpha_t \mathbf{x}_0 + \beta_t \mathbf{x}_1$ for functions $\alpha, \beta : [0, 1] \to \mathbb{R}$ such that $\alpha_0 = \beta_1 = 1$ and $\alpha_1 = \beta_0 = 0$, but more general choices of interpolating paths are possible.

We then define the stochastic interpolant between $\pi_0$ and $\pi_1$ to be the stochastic process $X = (X_t)_{t \in [0,1]}$ formed by sampling $X_0 \sim \pi_0$, $X_1 \sim \pi_1$ and $Z \sim \mathcal{N}(0, I_d)$ independently and setting $X_t = I(X_0, X_1, t) + \gamma_t Z$ for each $t \in (0, 1)$ where $\gamma : [0, 1] \to [0, \infty)$ is a function such that $\gamma_0 = \gamma_1 = 0$ and which determines the amount of Gaussian smoothing applied at time $t$. The motivation for including $\gamma_t Z$ is to smooth the interpolant marginals, as was originally explained by Albergo et al. (2023). Liu et al. (2023) and Albergo & Vanden-Eijnden (2023) omit $\gamma_t$, leading to deterministic paths between a given $X_0$ and $X_1$, while Albergo et al. (2023) and Lipman et al. (2023) work in the more general case.

The process $X$ is constructed so that its marginals deform smoothly from $\pi_0$ to $\pi_1$. However, $X$ is not a suitable choice of flow between $\pi_0$ and $\pi_1$ since it is not causal – it requires knowledge of $X_1$ to construct $X_t$. So, we seek a causal flow with the same marginals. The key insight is to define the expected velocity of $X$ by

$$v^X(\mathbf{x}, t) = \mathbb{E}\left[\dot{X}_t \mid X_t = \mathbf{x}\right], \quad \text{for all } \mathbf{x} \in \mathrm{supp}(X_t), \tag{2}$$

where $\dot{X}_t$ is the time derivative of $X_t$, and setting $v^X(\mathbf{x}, t) = 0$ for $\mathbf{x} \notin \mathrm{supp}(X_t)$ for each $t \in [0, 1]$. Then, the following result shows that $v^X(\mathbf{x}, t)$ generates a deterministic flow $Z$ between $\pi_0$ and $\pi_1$ with the same marginals as $X$ (Liu et al., 2023).

**Proposition 1.** *Suppose that $X$ is path-wise continuously differentiable, the expected velocity field $v^X(\mathbf{x}, t)$ exists and is locally bounded, and there exists a unique solution $Z^{\mathbf{x}}$ to (1) with velocity field $v^X$ for each $\mathbf{x} \in \mathbb{R}^d$. If $Z$ is the flow with velocity field $v^X(\mathbf{x}, t)$ starting in $\pi_0$, then $\mathrm{Law}(X_t) = \mathrm{Law}(Z_t)$ for all $t \in [0, 1]$.*

Sufficient conditions for $X$ to be path-wise continuously differentiable and for $v^X(\mathbf{x}, t)$ to exist and be locally bounded are that $\alpha$, $\beta$, and $\gamma$ are continuously differentiable and that $\pi_0, \pi_1$ have bounded support. We will assume that these conditions hold from now on. We also assume that all our data distributions are absolutely continuous with respect to the Lebesgue measure.

We can learn an approximation to $v^X$ by minimising the objective function

$$\mathcal{L}(v) = \int_0^1 \mathbb{E}\left[w_t \left\| v(X_t, t) - v^X(X_t, t) \right\|^2\right] \mathrm{d}t,$$

over all $v \in \mathcal{V}$ where $\mathcal{V}$ is some class of functions, for some weighting function $w_t : [0, 1] \to (0, \infty)$. (Typically, we will take $w_t = 1$ for all $t$.) As written, this objective is intractable, but we can show that

$$\mathcal{L}(v) = \int_0^1 \mathbb{E}\left[w_t \left\| v(X_t, t) - \dot{X}_t \right\|^2\right] \mathrm{d}t + \text{constant}, \tag{3}$$

where the constant is independent of $v$ (Lipman et al., 2023). This last integral can be empirically estimated in an unbiased fashion given access to samples $X_0 \sim \pi_0$, $X_1 \sim \pi_1$ and the functions $\alpha_t$, $\beta_t$, $\gamma_t$ and $w_t$. In practice, we often take $\mathcal{V}$ to be a class of functions parameterised by a neural network $v_\theta(\mathbf{x}, t)$ and minimise $\mathcal{L}(v_\theta)$ over the parameters $\theta$ using (3) and stochastic gradient descent. Our hope is that if our approximation $v_\theta$ is sufficiently close to $v^X$, and $Y$ is the flow with velocity field $v_\theta$, then if we take $Y_0 \sim \pi_0$ the distribution of $Y_1$ is approximately $\pi_1$.

Most frequently, flow matching is used as a generative modelling procedure. In order to model a distribution $\pi$ from which we have access to samples, we set $\pi_1 = \pi$ and take $\pi_0$ to be some reference distribution from which it is easy to sample, such as a standard Gaussian. Then, we use the flow matching procedure to learn an approximate flow $v_\theta(\mathbf{x}, t)$ between $\pi_0$ and $\pi_1$. We generate approximate samples from $\pi_1$ by sampling $Z_0 \sim \pi_0$ and integrating the flow equation (1) to find $Z_1$, which should be approximately distributed according to $\pi_1$.

## 3 Main results

We now present the main results of the paper. First, we show under three assumptions listed below that we can control the approximation error of the flow matching procedure under the 2-Wasserstein distance in terms of the quality of our approximation $v_\theta$. We obtain bounds that depend crucially on the spatial Lipschitz constant of the approximate flow. Second, we show how to control this Lipschitz constant for the true flow $v^X$ under a smoothness assumption on the data distributions, which is explained in Section 3.2. Combined with the first result, this will imply that for sufficiently regular $\pi_0$, $\pi_1$ there is a choice of $\mathcal{V}$ which contains the true flow such that, if we optimise $L(v)$ over all $v \in \mathcal{V}$, then we can bound the error of the flow matching procedure. Additionally, our bound will be polynomial in the $L^2$ approximation error. The results of this section will be proved under the following three assumptions.

**Assumption 1** (Bound on $L^2$ approximation error)**.** *The true and approximate drifts $v^X(\mathbf{x}, t)$ and $v_\theta(\mathbf{x}, t)$ satisfy $\int_0^1 \mathbb{E}\left[\|v_\theta(X_t, t) - v^X(X_t, t)\|^2\right] \mathrm{d}t \leq \varepsilon^2$.*

**Assumption 2** (Existence and uniqueness of smooth flows)**.** *For each $\mathbf{x} \in \mathbb{R}^d$ and $s \in [0, 1]$ there exist unique flows $(Y_{s,t}^{\mathbf{x}})_{t \in [s,1]}$ and $(Z_{s,t}^{\mathbf{x}})_{t \in [s,1]}$ starting in $Y_{s,s}^{\mathbf{x}} = \mathbf{x}$ and $Z_{s,s}^{\mathbf{x}} = \mathbf{x}$ with velocity fields $v_\theta(\mathbf{x}, t)$ and $v^X(\mathbf{x}, t)$ respectively. Moreover, $Y_{s,t}^{\mathbf{x}}$ and $Z_{s,t}^{\mathbf{x}}$ are continuously differentiable in $\mathbf{x}$, $s$ and $t$.*

**Assumption 3** (Regularity of approximate velocity field)**.** *The approximate flow $v_\theta(\mathbf{x}, t)$ is differentiable in both inputs. Also, for each $t \in (0, 1)$ there is a constant $L_t$ such that $v_\theta(\mathbf{x}, t)$ is $L_t$-Lipschitz in $\mathbf{x}$.*

Assumption 1 is the natural assumption on the training error given we are learning with the $L^2$ training loss in (3). Assumption 2 is required since to perform flow matching we need to be able to solve the ODE (1). Without a smoothness assumption on $v_\theta(\mathbf{x}, t)$, it would be possible for the marginals $Y_t$ of the solution to (1) initialised in $Y_0 \sim \pi_0$ to quickly concentrate on subsets of $\mathbb{R}^d$ of arbitrarily small measure under the distribution of $X_t$. Then, there would be choices of $v_\theta(\mathbf{x}, t)$ which were very different from $v^X(\mathbf{x}, t)$ on these sets of small measure while being equal to $v^X(\mathbf{x}, t)$ everywhere else. For these $v_\theta(\mathbf{x}, t)$ the $L^2$ approximation error can be kept arbitrarily small while the error of the flow matching procedure is made large. We therefore require some smoothness assumption on $v_\theta(\mathbf{x}, t)$ – we choose to make Assumption 3 since we can show that it holds for the true velocity field, as we do in Section 3.2.

### 3.1 Controlling the Wasserstein difference between flows

Our first main result is the following, which bounds the error of the flow matching procedure in 2-Wasserstein distance in terms of the $L^2$ approximation error and the Lipschitz constant of the approximate flow.

**Theorem 1.** *Suppose that $\pi_0$, $\pi_1$ are probability distributions on $\mathbb{R}^d$, that $Y$ is the flow starting in $\pi_0$ with velocity field $v_\theta$, and $\hat{\pi}_1$ is the law of $Y_1$. Then, under Assumptions 1-3, we have*

$$W_2(\hat{\pi}_1, \pi_1) \leq \varepsilon \exp\left\{\int_0^1 L_t \, \mathrm{d}t\right\}.$$

We see that the error depends linearly on the $L^2$ approximation error $\varepsilon$ and exponentially on the integral of the Lipschitz constant of the approximate flow $L_t$. Ostensibly, the exponential dependence on $\int_0^1 L_t \, \mathrm{d}t$

is undesirable, since $v_\theta$ may have a large spatial Lipschitz constant. However, we will show in Section 3.2 that the true velocity field is $L_t^*$-Lipschitz for a choice of $L_t^*$ such that $\int_0^1 L_t^* \, dt$ depends logarithmically on the amount of Gaussian smoothing. Thus, we may optimise (3) over a class of functions $\mathcal{V}$ which are all $L_t^*$-Lipschitz, knowing that this class contains the true velocity field, and that if $v_\theta$ lies in this class we get a bound on the approximation error which is polynomial in the $L^2$ approximation error, providing we make a suitable choice of $\alpha_t$, $\beta_t$ and $\gamma_t$.

We remark that our Theorem 1 is similar in content to Proposition 3 in Albergo & Vanden-Eijnden (2023). The crucial difference is that we work with a time-varying Lipschitz constant, which is required in practice since in many cases $L_t$ explodes as $t \to 0$ or $1$, as happens for example if $\pi_0$ or $\pi_1$ do not have full support.

A key ingredient in the proof of Theorem 1 will be the Alekseev–Gröbner formula, which provides a way to control the difference between the solutions to two different ODEs in terms of the difference between their drifts (Gröbner, 1960; Alekseev, 1961).

**Proposition 2** (Alekseev–Gröbner). *Let $\mu(\mathbf{x}, t) : \mathbb{R}^d \times [0, T] \to \mathbb{R}^d$ be a function which is continuous in $t$ and continuously differentiable in $\mathbf{x}$. Let $X : \mathbb{R}^d \times [0, T]^2 \to \mathbb{R}^d$ be a continuous solution of the integral equation*

$$X_{s,t}^{\mathbf{x}} = \mathbf{x} + \int_s^t \mu(r, X_{s,r}^{\mathbf{x}}) \, dr,$$

*and let $Y : [0, T] \to \mathbb{R}^d$ be another continuously differentiable process. Then*

$$X_{0,T}^{Y_0} - Y_T = \int_0^T \left( \nabla_{\mathbf{x}} X_{r,T}^{Y_r} \right) \left( \mu(r, Y_r) - \frac{dY_r}{dt} \right) dr.$$

We now give the proof of Theorem 1.

*Proof of Theorem 1.* Define $Y_{s,t}^{\mathbf{x}}$ and $Z_{s,t}^{\mathbf{x}}$ as in Assumption 2. As $Y_{s,t}^{\mathbf{x}} \in C(\mathbb{R}^d \times [0,1]^2)$, $Z_{0,t}^{\mathbf{x}} \in C^1(\mathbb{R}^d \times [0,1])$ and $v_\theta(\mathbf{x}, t) \in C^{0,1}(\mathbb{R}^d \times [0,1])$ for any $\mathbf{x} \in \mathbb{R}^d$, the Alekseev–Gröbner formula implies that

$$Y_{0,t}^{\mathbf{x}} - Z_{0,t}^{\mathbf{x}} = \int_0^t \left( \nabla_{\mathbf{x}} Y_{s,t}^{Z_s} \right) \left( v_\theta(Z_s, s) - v^X(Z_s, s) \right) ds. \tag{4}$$

We can write $Y_{s,t}^{\mathbf{x}} = \mathbf{x} + \int_s^t v_\theta(Y_{s,r}^{\mathbf{x}}, r) \, dr$, and so

$$\nabla_{\mathbf{x}} Y_{s,t}^{\mathbf{x}} = I + \int_s^t \nabla_{\mathbf{x}} v_\theta(Y_{s,r}^{\mathbf{x}}, r) \nabla_{\mathbf{x}} Y_{s,r}^{\mathbf{x}} \, dr.$$

It follows that

$$\frac{\partial}{\partial t} \left\| \nabla_{\mathbf{x}} Y_{s,t}^{\mathbf{x}} \right\|_{\mathrm{op}} \leq \left\| \frac{\partial}{\partial t} \nabla_{\mathbf{x}} Y_{s,t}^{\mathbf{x}} \right\|_{\mathrm{op}} = \left\| \nabla_{\mathbf{x}} v_\theta(Y_{s,t}^{\mathbf{x}}, t) \nabla_{\mathbf{x}} Y_{s,t}^{\mathbf{x}} \right\|_{\mathrm{op}} \leq L_t \left\| \nabla_{\mathbf{x}} Y_{s,t}^{\mathbf{x}} \right\|_{\mathrm{op}},$$

where $\| \cdot \|_{\mathrm{op}}$ denotes the operator norm with respect to the $\ell^2$ metric on $\mathbb{R}^d$. Therefore, by Grönwall's lemma we have $\left\| \nabla_{\mathbf{x}} Y_{s,t}^{\mathbf{x}} \right\|_{\mathrm{op}} \leq \exp \left\{ \int_s^t L_r \, dr \right\} \leq K$, where $K := \exp \left\{ \int_0^1 L_t \, dt \right\}$. Applied to (4) at $t = 1$, we get

$$\left\| Y_1^{\mathbf{x}} - Z_1^{\mathbf{x}} \right\|_2 \leq \int_0^1 \left\| \nabla_{\mathbf{x}} Y_{s,1}^{Z_s} \right\|_{\mathrm{op}} \left\| v^X(Z_s, s) - v_\theta(Z_s, s) \right\|_2 ds \leq K \int_0^1 \left\| v^X(Z_s, s) - v_\theta(Z_s, s) \right\|_2 ds.$$

Letting $\mathbf{x} \sim \pi_0$ and taking expectations, we deduce

$$\mathbb{E} \left[ \| Y_1 - Z_1 \|_2^2 \right] \leq K^2 \mathbb{E} \left[ \left( \int_0^1 \left\| v^X(Z_t, t) - v_\theta(Z_t, t) \right\|_2 dt \right)^2 \right]$$

$$\leq K^2 \int_0^1 \mathbb{E} \left[ \left\| v_\theta(X_t, t) - v^X(X_t, t) \right\|^2 \right] dt.$$

Since $W_2(\hat{\pi}_1, \pi_1) = W_2(\mathrm{Law}(Y_1), \mathrm{Law}(Z_1)) \leq \mathbb{E} \left[ \| Y_1 - Z_1 \|_2^2 \right]^{1/2}$, the result follows from our assumption on the $L^2$ error and the definition of $K$. $\qquad \square$

The application of the Alekseev–Gröbner formula in (4) is not symmetric in $Y$ and $Z$. Applying the formula with the roles of $Y$ and $Z$ reversed would mean we require a bound on $\left\|\nabla_{\mathbf{x}} Y_{s,t}^{\mathbf{x}}\right\|_{\mathrm{op}}$, which we could bound in terms of the Lipschitz constant of the true velocity field, leading to a more natural condition. However, we would then also be required to control the velocity approximation error $\|v_\theta(Y_t, t) - v^X(Y_t, t)\|_2$ along the paths of $Y$, which we have much less control over due to the nature of the training objective.

## 3.2 Smoothness of the velocity fields

As remarked in Section 3.1, the bound in Theorem 1 is exponentially dependent on the Lipschitz constant of $v_\theta$. In this section, we control the corresponding Lipschitz constant of $v^X$. We prefer this to controlling the Lipschitz constant of $v_\theta$ directly since this is determined by our choice of $\mathcal{V}$, the class of functions over which we optimise (3).

In this section, we will relax the constraints from Section 2 that $\gamma_0 = \gamma_1 = 0$ and $\alpha_0 = \beta_1 = 1$. We do this because allowing $\gamma_t > 0$ for all $t \in [0, 1]$ means that we have some Gaussian smoothing at every $t$, which will help ensure the resulting velocity fields are sufficiently regular. This will mean that instead of learning a flow between $\pi_0$ and $\pi_1$ we will actually be learning a flow between $\tilde{\pi}_0$ and $\tilde{\pi}_1$, the distributions of $\alpha_0 X_0 + \gamma_0 Z$ and $\beta_1 X_1 + \gamma_1 Z$ respectively. However, if we centre $X_0$ and $X_1$ so that $\mathbb{E}[X_0] = \mathbb{E}[X_1] = 0$ and let $R$ be such that $\|X_0\|_2, \|X_1\|_2 \leq R$, then $W_2(\tilde{\pi}_0, \pi_0)^2 \leq (1 - \alpha_0)^2 R^2 + d\gamma_0^2$ and similarly for $\tilde{\pi}_1, \pi_1$, so it follows that by taking $\gamma_0, \gamma_1$ sufficiently close to 0 and $\alpha_0, \beta_1$ sufficiently close to 1 we can make $\tilde{\pi}_0, \tilde{\pi}_1$ very close in 2-Wasserstein distance to $\pi_0, \pi_1$ respectively. Note that if $\pi_0$ is easy to sample from then $\tilde{\pi}_0$ is also easy to sample from (we may simulate samples from $\tilde{\pi}_0$ by drawing $X_0 \sim \pi_0$, $Z \sim \mathcal{N}(0, I_d)$ independently, setting $\tilde{X}_0 = \alpha_0 X_0 + \gamma_0 Z$ and noting that $\tilde{X}_0 \sim \tilde{\pi}_0$), so we can run the flow matching procedure starting from $\tilde{\pi}_0$.

For arbitrary choices of $\pi_0$ and $\pi_1$ the expected velocity field $v^X(\mathbf{x}, t)$ can be very badly behaved, and so we will require an additional regularity assumption on the process $X$. This notion of regularity is somewhat non-standard, but is the natural one emerging from our proofs and bears some similarity to quantities controlled in stochastic localization (see discussion below).

**Definition 1.** *For a real number $\lambda \geq 1$, we say an $\mathbb{R}^d$-valued random variable $W$ is $\lambda$-**regular** if, whenever we take $\tau \in (0, \infty)$ and $\xi \sim \mathcal{N}(0, \tau^2 I_d)$ independently of $W$ and set $W' = W + \xi$, for all $\mathbf{x} \in \mathbb{R}^d$ we have*

$$\left\|\mathrm{Cov}_{\xi|W'=\mathbf{x}}(\xi)\right\|_{\mathrm{op}} \leq \lambda \tau^2.$$

*We say that a distribution on $\mathbb{R}^d$ is $\lambda$-regular if the associated random variable is $\lambda$-regular.*

**Assumption 4** (Regularity of data distributions). *For some $\lambda \geq 1$, $\alpha_t X_0 + \beta_t X_1$ is $\lambda$-regular for all $t \in [0, 1]$.*

To understand what it means for a random variable $W$ to be $\lambda$-regular, note that before conditioning on $W'$, we have $\|\mathrm{Cov}(\xi)\|_{\mathrm{op}} = \tau^2$. We can think of conditioning on $W' = \mathbf{x}$ as re-weighting the distribution of $\xi$ proportionally to $p_W(\mathbf{x} - \xi)$, where $p_W(\cdot)$ is the density function of $W$. So, $W$ is $\lambda$-regular if this re-weighting causes the covariance of $\xi$ to increase by at most a factor of $\lambda$ for any choice of $\tau$.

Informally, if $\tau$ is much smaller than the scale over which $\log p_W(\cdot)$ varies then this re-weighting should have negligible effect, while for large $\tau$ we can write $\|\mathrm{Cov}_{\xi|W'=\mathbf{x}}(\xi)\|_{\mathrm{op}} = \|\mathrm{Cov}_{W|W'=\mathbf{x}}(W)\|_{\mathrm{op}}$ and we expect this to be less than $\tau^2$ once $\tau$ is much greater than the typical magnitude of $W$. Thus $\lambda \approx 1$ should suffice for sufficiently small or large $\tau$ and the condition that $W$ is $\lambda$-regular controls how much conditioning on $W'$ can change the behavior of $\xi$ as we transition between these two extremes.

If $W$ is log-concave or alternatively Gaussian on a linear subspace of $\mathbb{R}^d$, then we show in Appendix A.1 that $W$ is $\lambda$-regular with $\lambda = 1$. Additionally, we also show that a mixture of Gaussians $\pi = \sum_{i=1}^{K} \mu_i \mathcal{N}(\mathbf{x}_i, \sigma^2 I_d)$, where the weights $\mu_i$ satisfy $\sum_{i=1}^{K} \mu_i = 1$, $\sigma > 0$, and $\|\mathbf{x}_i\| \leq R$ for $i = 1, \ldots, K$, is $\lambda$-regular with $\lambda = 1 + (R^2/\sigma^2)$. This shows that $\lambda$-regularity can hold even for highly multimodal distributions. More generally, we show in Appendix A.2 that we always have $\|\mathrm{Cov}_{\xi|W'=\mathbf{x}}(\xi)_{\mathrm{op}}\| \leq O(d\tau^2)$ with high probability. Then, we can interpret Definition 1 as insisting that this inequality always holds, where the dependence on $d$ is incorporated into the parameter $\lambda$. (We see that in the worst case $\lambda$ may scale linearly with $d$, but we expect in practice that $\lambda$ will be approximately constant in many cases of interest.)

Controlling covariances of random variables given noisy observations of those variables is also a problem which arises in stochastic localization (Eldan, 2013; 2020), and similar bounds to the ones we use here have been established in this context. For example, Alaoui & Montanari (2022) show that $\mathbb{E}\left[\mathrm{Cov}_{\xi|W'=\mathbf{x}}(W')\right] \leq \tau^2 I_d$ in our notation. However, the bounds from stochastic localization only hold in expectation over $\mathbf{x}$ distributed according to the law of $W'$, whereas we require bounds that hold pointwise, or at least for $\mathbf{x}$ distributed according to the law of $Y_{s,t}^{\mathbf{x}}$, which is much harder to control.

We now aim to bound the Lipschitz constant of the true velocity field under Assumption 4. The first step is to show that $v^X$ is differentiable and to get an explicit formula for its derivative. In the following sections, we abbreviate the expectation and covariance conditioned on $X_t = \mathbf{x}$ to $\mathbb{E}_{\mathbf{x}}[\,\cdot\,]$ and $\mathrm{Cov}_{\mathbf{x}}(\cdot)$ respectively.

**Lemma 1.** *If $X$ is the stochastic interpolant between $\pi_0$ and $\pi_1$, then $v^X(\mathbf{x}, t)$ is differentiable with respect to $\mathbf{x}$ and*

$$\nabla_{\mathbf{x}} v^X(\mathbf{x}, t) = \frac{\dot{\gamma}_t}{\gamma_t} I_d - \frac{1}{\gamma_t} \mathrm{Cov}_{\mathbf{x}}(\dot{X}_t, Z).$$

The proof of Lemma 1 is given in Appendix B.1. Using Lemma 1, we can derive the following theorem, which provides a bound on the time integral of the Lipschitz constant of $v_\theta$.

**Theorem 2.** *If $X$ is the stochastic interpolant between two distributions $\pi_0$ and $\pi_1$ on $\mathbb{R}^d$ with bounded support, then under Assumption 4 for each $t \in (0, 1)$ there is a constant $L_t^*$ such that $v^X(\mathbf{x}, t)$ is $L_t^*$-Lipschitz in $\mathbf{x}$ and*

$$\int_0^1 L_t^* \, \mathrm{d}t \leq \lambda \left( \int_0^1 \frac{|\dot{\gamma}_t|}{\gamma_t} \, \mathrm{d}t \right) + \lambda^{1/2} R \left( \int_0^1 \frac{|\dot{\alpha}_t|}{\gamma_t} \, \mathrm{d}t + \int_0^1 \frac{|\dot{\beta}_t|}{\gamma_t} \, \mathrm{d}t \right),$$

*where* $\mathrm{supp}\,\pi_0,\ \mathrm{supp}\,\pi_1 \subseteq \bar{B}(0, R)$, *the closed ball of radius $R$ around the origin.*

*Proof.* Using Lemma 1 plus the fact that $\dot{X}_t = \dot{\alpha}_t X_0 + \dot{\beta}_t X_1 + \dot{\gamma}_t Z$, we can write

$$\nabla_{\mathbf{x}} v^X(\mathbf{x}, t) = \frac{\dot{\gamma}_t}{\gamma_t}\Big( I_d - \mathrm{Cov}_{\mathbf{x}}(Z, Z) \Big) - \frac{\dot{\alpha}_t}{\gamma_t} \mathrm{Cov}_{\mathbf{x}}(X_0, Z) - \frac{\dot{\beta}_t}{\gamma_t} \mathrm{Cov}_{\mathbf{x}}(X_1, Z).$$

Since we can express $X_t = (\alpha_t X_0 + \beta_t X_1) + \gamma_t Z$ where $\alpha_t X_0 + \beta_t X_1$ is $\lambda$-regular by Assumption 4 and $\gamma_t Z \sim \mathcal{N}(0, \gamma_t^2 I_d)$, we have $\|\mathrm{Cov}_{\mathbf{x}}(\gamma_t Z)\|_{\mathrm{op}} \leq \lambda \gamma_t^2$. It follows that

$$\|I_d - \mathrm{Cov}_{\mathbf{x}}(Z, Z)\|_{\mathrm{op}} \leq \max\left(1, \|\mathrm{Cov}_{\mathbf{x}}(Z, Z)\|_{\mathrm{op}}\right) \leq \lambda.$$

Also, using Cauchy–Schwarz we have

$$\|\mathrm{Cov}_{\mathbf{x}}(X_0, Z)\|_{\mathrm{op}} \leq \|\mathrm{Cov}_{\mathbf{x}}(X_0)\|_{\mathrm{op}}^{1/2} \|\mathrm{Cov}_{\mathbf{x}}(Z)\|_{\mathrm{op}}^{1/2} \leq \lambda^{1/2} R,$$

and a similar result holds for $X_1$ in place of $X_0$.

Putting this together, we get

$$\left\|\nabla_{\mathbf{x}} v^X(\mathbf{x}, t)\right\|_{\mathrm{op}} \leq \frac{|\dot{\gamma}_t|}{\gamma_t} \|I_d - \mathrm{Cov}_{\mathbf{x}}(Z, Z)\|_{\mathrm{op}} + \frac{|\dot{\alpha}_t|}{\gamma_t} \|\mathrm{Cov}_{\mathbf{x}}(X_0, Z)\|_{\mathrm{op}} + \frac{|\dot{\beta}_t|}{\gamma_t} \|\mathrm{Cov}_{\mathbf{x}}(X_1, Z)\|_{\mathrm{op}}$$

$$\leq \lambda \frac{|\dot{\gamma}_t|}{\gamma_t} + \lambda^{1/2} R \left( \frac{|\dot{\alpha}_t|}{\gamma_t} + \frac{|\dot{\beta}_t|}{\gamma_t} \right). \tag{5}$$

Finally, since $v^X(\mathbf{x}, t)$ is differentiable with uniformly bounded derivative, it follows that $v^X(\mathbf{x}, t)$ is $L_t^*$-Lipschitz with $L_t^* = \sup_{\mathbf{x} \in \mathbb{R}^d} \left\|\nabla_{\mathbf{x}} v^X(\mathbf{x}, t)\right\|_{\mathrm{op}}$. Integrating (5) from $t = 0$ to $t = 1$, the result follows. $\qquad \square$

We now provide some intuition for the bound in Theorem 2. The key term on the RHS is the first one, which we typically expect to be on the order of $\log(\gamma_{\max}/\gamma_{\min})$ where $\gamma_{\max}$ and $\gamma_{\min}$ are the maximum and minimum values taken by $\gamma_t$ on the interval $[0, 1]$ respectively. This is suggested by the following lemma.

**Lemma 2.** *Suppose that $\gamma_0 = \gamma_1 = \gamma_{\min}$ and $\gamma_t$ increases smoothly from $\gamma_{\min}$ at $t = 0$ to $\gamma_{\max}$ before decreasing back to $\gamma_{\min}$ at $t = 1$. Then $\int_0^1 (|\dot{\gamma}_t|/\gamma_t) \, \mathrm{d}t = 2\log(\gamma_{\max}/\gamma_{\min})$.*

*Proof.* Note that we can write $|\dot{\gamma}_t|/\gamma_t = |\mathrm{d}(\log \gamma_t)/\mathrm{d}t|$, so $\int_0^1 (|\dot{\gamma}_t|/\gamma_t)\,\mathrm{d}t$ is simply the total variation of $\log \gamma_t$ over the interval $[0, 1]$. By the assumptions on $\gamma_t$, we see this total variation is equal to $2\log(\gamma_{\max}/\gamma_{\min})$. □

The first term on the RHS in Theorem 2 also explains the need to relax the boundary conditions, since $\gamma_0 = 0$ or $\gamma_1 = 0$ would cause $\int_0^1 (|\dot{\gamma}_t|/\gamma_t)\,\mathrm{d}t$ to diverge.

We can ensure the second terms in Theorem 2 are relatively small through a suitable choice of $\alpha_t$, $\beta_t$ and $\gamma_t$. Since $\alpha_t$ and $\beta_t$ are continuously differentiable, $|\dot{\alpha}_t|$ and $|\dot{\beta}_t|$ are bounded, so to control these terms we should pick $\gamma_t$ such that $R\int_0^1 (1/\gamma_t)\,\mathrm{d}t$ is small, while respecting the fact that we need $\gamma_t \ll 1$ at the boundaries. One sensible choice among many would be $\gamma_t = 2R\sqrt{(\delta+t)(1+\delta-t)}$ for some $\delta \ll 1$, where $\int_0^1 (|\dot{\gamma}_t|/\gamma_t)\,\mathrm{d}t \approx 2\log(1/\sqrt{\delta})$ and $\int_0^1 (1/\gamma_t)\,\mathrm{d}t \approx 2\pi$. In this case, the bound of Theorem 2 implies $\int_0^1 L_t^*\,\mathrm{d}t \leq \lambda \log(1/\delta) + 2\pi\lambda^{1/2}\sup_{t\in[0,1]}(|\dot{\alpha}_t| + |\dot{\beta}_t|)$, so if $\delta \ll 1$ the dominant term in our bound is $\lambda\log(1/\delta)$.

### 3.3 Bound on the Wasserstein error of flow matching

Now, we demonstrate how to combine the results of the previous two sections to get a bound on the error of the flow matching procedure that applies in settings of practical interest. In Section 3.2, we saw that we typically have $\int_0^1 L_t^*\,\mathrm{d}t \sim \lambda\log(\gamma_{\max}/\gamma_{\min})$. The combination of this logarithm and the exponential in Theorem 1 should give us bounds on the 2-Wasserstein error which are polynomial in $\varepsilon$ and $\gamma_{\max}, \gamma_{\min}$.

The main idea is that in order to ensure that we may apply Theorem 1, we should optimise $\mathcal{L}(v)$ over a class of functions $\mathcal{V}$ which all satisfy Assumption 3. (Technically, we only need Assumption 3 to hold at the minimum of $\mathcal{L}(v)$, but since it is not possible to know what this minimum will be ex ante, it is easier in practice to enforce that Assumption 3 holds for all $v \in \mathcal{V}$.) Given distributions $\pi_0, \pi_1$ satisfying Assumption 4 as well as specific choices of $\alpha_t$, $\beta_t$ and $\gamma_t$ we define $K_t = \lambda(|\dot{\gamma}_t|/\gamma_t) + \lambda^{1/2}R\{(|\dot{\alpha}_t|/\alpha_t) + (|\dot{\beta}_t|/\beta_t)\}$ and let $\mathcal{V}$ be the set of functions $v : \mathbb{R}^d \times [0,1] \to \mathbb{R}^d$ which are $K_t$-Lipschitz in $\mathbf{x}$ for all $t \in [0,1]$.

**Theorem 3.** *Suppose that Assumptions 1-4 hold and that $\gamma_t$ is concave on $[0,1]$. Then $v^X \in \mathcal{V}$ and for any $v_\theta \in \mathcal{V}$, if $Y$ is a flow starting in $\tilde{\pi}_0$ with velocity field $v_\theta$ and $\hat{\pi}_1$ is the law of $Y_1$,*

$$W_2(\hat{\pi}_1, \tilde{\pi}_1) \leq C^{\lambda^{1/2}}\varepsilon\left(\frac{\gamma_{\max}}{\gamma_{\min}}\right)^{2\lambda},$$

*where $\gamma_{\min} = \inf_{t\in[0,1]}\gamma_t$, $\gamma_{\max} = \sup_{t\in[0,1]}\gamma_t$ and $C = \exp\left\{R\left\{\int_0^1 (|\dot{\alpha}_t|/\gamma_t)\,\mathrm{d}t + \int_0^1 (|\dot{\beta}_t|/\gamma_t)\,\mathrm{d}t\right\}\right\}$.*

*Proof.* First, note that $v^X$ is $K_t$-Lipschitz in $\mathbf{x}$ for all $t \in [0,1]$ by the proof of Theorem 2, so $v^X \in \mathcal{V}$. Then, since $\gamma_t$ is concave, $\int_0^1 (|\dot{\gamma}_t|/\gamma_t)\,\mathrm{d}t \leq 2\log(\gamma_{\max}/\gamma_{\min})$ by Lemma 2. Thus, $\int_0^1 K_t\,\mathrm{d}t \leq 2\lambda\log(\gamma_{\max}/\gamma_{\min}) + \lambda^{1/2}\log C$. Hence for any $v_\theta \in \mathcal{V}$ we may apply Theorem 1 to get

$$W_2(\hat{\pi}_1, \tilde{\pi}_1) \leq \varepsilon\exp\left\{\int_0^1 K_t\,\mathrm{d}t\right\} \leq C^{\lambda^{1/2}}\varepsilon\left(\frac{\gamma_{\max}}{\gamma_{\min}}\right)^{2\lambda},$$

where $\tilde{\pi}_0, \tilde{\pi}_1$ are replacing $\pi_0, \pi_1$ since we have relaxed the boundary conditions. □

Theorem 3 shows that if we optimise $\mathcal{L}(v)$ over the class $\mathcal{V}$, we can bound the resulting distance between the flow matching paths. We typically choose $\gamma_t$ to scale with $R$, so $C$ should be thought of as a constant that is $\Theta(1)$ and depends only on the smoothness of the interpolating paths. For a given distribution, $\lambda$ is fixed and our bound becomes polynomial in $\varepsilon$ and $\gamma_{\max}/\gamma_{\min}$.

Finally, our choice of $\gamma_{\min}$ controls how close the distributions $\tilde{\pi}_0$, $\tilde{\pi}_1$ are to $\pi_0$, $\pi_1$ respectively. We can combine this with our bound on $W_2(\hat{\pi}_1, \tilde{\pi}_1)$ to get the following result.

**Theorem 4.** *Suppose that Assumptions 1-4 hold, that $\gamma_t$ is concave on $[0,1]$, and that $\alpha_0 = \beta_1 = 1$ and $\gamma_0 = \gamma_1 = \gamma_{\min}$. Then, for any $v_\theta \in \mathcal{V}$, if $Y$ is a flow starting in $\tilde{\pi}_0$ with velocity field $v_\theta$ and $\hat{\pi}_1$ is the law of $Y_1$,*

$$W_2(\hat{\pi}_1, \pi_1) \leq C^{\lambda^{1/2}}\varepsilon\left(\frac{\gamma_{\max}}{\gamma_{\min}}\right)^{2\lambda} + \sqrt{d}\gamma_{\min}$$

*where* $\gamma_{\min}$, $\gamma_{\max}$ *and* $C$ *are as before.*

*Proof.* We have $W_2(\hat{\pi}_1, \pi_1) \leq W_2(\hat{\pi}_1, \tilde{\pi}_1) + W_2(\tilde{\pi}_1, \pi_1)$. The first term is controlled by Theorem 3, while for the second term we have $W_2(\tilde{\pi}_1, \pi_1)^2 \leq (1 - \beta_1)R^2 + d\gamma_0^2$, as noted previously. Using $\beta_1 = 1$ and combining these two results completes the proof. $\qquad\square$

Optimizing the expression in Theorem 4 over $\gamma_{\min}$, we see that for a given $L^2$ error tolerance $\varepsilon$, we should take $\gamma_{\min} \sim d^{-1/(4\lambda+2)}\varepsilon^{1/(2\lambda+1)}$. We deduce the following corollary.

**Corollary 1.** *In the setting of Theorem 4, if we take* $\gamma_{\min} \sim d^{-1/(4\lambda+2)}\varepsilon^{1/(2\lambda+1)}$ *then the total Wasserstein error of the flow matching procedure is of order* $W_2(\hat{\pi}_1, \pi_1) \lesssim d^{2\lambda/(4\lambda+2)}\varepsilon^{1/(2\lambda+1)}$.

## 4  Application to probability flow ODEs

For generative modelling applications, $\pi_1$ is the target distribution and $\pi_0$ is typically chosen to be a simple reference distribution. A common practical choice for $\pi_0$ is a standard Gaussian distribution, and in this setting the flow matching framework reduces to the probability flow ODE (PF-ODE) framework for diffusion models (Song et al., 2021b). (Technically, this corresponds to running the PF-ODE framework for infinite time. Finite time versions of the PF-ODE framework can be recovered by taking $\beta_0$ to be positive but small.) Previously, this correspondence has been presented by taking $\gamma_t = 0$ for all $t \in [0,1]$ and $\pi_0 = \mathcal{N}(0, I_d)$ in our notation from Section 2 (Liu et al., 2023). Instead, we choose to set $\alpha_t = 0$ for all $t \in [0,1]$ and have $\gamma_t > 0$, which recovers exactly the same framework, just with $Z$ playing the role of the reference random variable rather than $X_0$. We do this because it allows us to apply the results of Section 3 directly.

Because we have this alternative representation, we can strengthen our results in the PF-ODE setting. First, note that Assumption 4 simplifies, so that we only need to assume that $\pi_1$ is $\lambda$-regular. We thus replace Assumption 4 with Assumption 4' for the rest of this section.

**Assumption 4'** (Regularity of data distribution, Gaussian case)**.** *For some* $\lambda \geq 1$, *the distribution* $\pi_1$ *is* $\lambda$-*regular.*

We also get the following alternative form of Lemma 1 in this setting, which is proved in Appendix B.2.

**Lemma 3.** *If* $X$ *is the stochastic interpolant in the PF-ODE setting above, then* $v^X(\mathbf{x}, t)$ *is differentiable with respect to* $\mathbf{x}$ *and*

$$\nabla_{\mathbf{x}} v^X(\mathbf{x}, t) = \frac{\dot{\gamma}_t}{\gamma_t} I_d - \left( \frac{\dot{\gamma}_t}{\gamma_t} - \frac{\dot{\beta}_t}{\beta_t} \right) \mathrm{Cov}_{\mathbf{x}}(Z).$$

Using Lemma 3, we can follow a similar argument to the proof of Theorem 2 to get additional control on the terms $L_t^*$.

**Theorem 5.** *If* $X$ *is the stochastic interpolant in the PF-ODE setting above, then under Assumption 4' for each* $t \in (0,1)$ *there is a constant* $L_t^*$ *such that* $v^X(\mathbf{x}, t)$ *is* $L_t^*$-*Lipschitz in* $\mathbf{x}$ *and*

$$\int_0^1 L_t^* \, \mathrm{d}t \leq \lambda \left( \int_0^1 \frac{|\dot{\gamma}_t|}{\gamma_t} \, \mathrm{d}t \right) + \int_0^1 \min \left\{ \lambda \frac{|\dot{\beta}_t|}{\beta_t}, \ \lambda^{1/2} R \, \frac{|\dot{\beta}_t|}{\gamma_t} \right\} \, \mathrm{d}t.$$

*Proof.* From Theorem 2 we know that $v^X(\mathbf{x}, t)$ is differentiable and Lipschitz with some constant $L_t^*$. From (5) in the proof of Theorem 2, for any $\mathbf{x} \in \mathbb{R}^d$ we have

$$\left\| \nabla_{\mathbf{x}} v^X(\mathbf{x}, t) \right\|_{\mathrm{op}} \leq \lambda \frac{|\dot{\gamma}_t|}{\gamma_t} + \lambda^{1/2} R \, \frac{|\dot{\beta}_t|}{\gamma_t}, \tag{6}$$

since $\dot{\alpha}_t = 0$ in this setting. Also, using Lemma 3,

$$\left\| \nabla_{\mathbf{x}} v^X(\mathbf{x}, t) \right\|_{\mathrm{op}} \leq \frac{|\dot{\gamma}_t|}{\gamma_t} \left\| I_d - \mathrm{Cov}_{\mathbf{x}}(Z) \right\|_{\mathrm{op}} + \frac{|\dot{\beta}_t|}{\beta_t} \left\| \mathrm{Cov}_{\mathbf{x}}(Z) \right\|_{\mathrm{op}} \leq \lambda \frac{|\dot{\gamma}_t|}{\gamma_t} + \lambda \frac{|\dot{\beta}_t|}{\beta_t}. \tag{7}$$

As $L_t^* = \sup_{\mathbf{x} \in \mathbb{R}^d} \left\| \nabla_{\mathbf{x}} v^X(\mathbf{x}, t) \right\|_{\mathrm{op}}$, combining (6), (7) and integrating from $t = 0$ to $t = 1$ gives the result. $\quad\square$

To interpret Theorem 5, recall that the boundary conditions we are operating under are $\gamma_0 = \beta_1 = 1$, $\beta_0 = 0$ and $\gamma_1 \ll 1$, so that $\tilde{\pi}_0 = \mathcal{N}(0, I_d)$ and $\tilde{\pi}_1$ is $\pi_1$ plus a small amount of Gaussian noise at scale $\gamma_1$. The following corollary gives the implications of Theorem 5 in the standard variance-preserving (VP) and variance-exploding (VE) PF-ODE settings of Song et al. (2021b).

**Corollary 2.** *Suppose that we are in the setting of Theorem 5, so that $v^X(\mathbf{x}, t)$ is $L_t^*$-Lipschitz in $\mathbf{x}$. Then,*

> *(i) if $\gamma_t = R\cos((\frac{\pi}{2} - \delta)t)$ and $\beta_t = \sin((\frac{\pi}{2} - \delta)t)$ for $\delta \ll 1$, corresponding to the VP ODE framework of Song et al. (2021b), then $\int_0^1 L_t^* \, \mathrm{d}t \leq \lambda(1 + \log(1/\gamma_1))$;*

> *(ii) if $\beta_t = 1$ for all $t \in [0, 1]$ and $\gamma_t$ is decreasing, corresponding to the VE ODE framework of Song et al. (2021b), then $\int_0^1 L_t^* \, \mathrm{d}t \leq \lambda \log(1/\gamma_1)$.*

*Proof.* In both cases, we apply Theorem 5 to bound $\int_0^1 L_t^*$. As $\gamma_t$ is decreasing, we have $\int_0^1 |\dot{\gamma}_t|/\gamma_t \, \mathrm{d}t = \log(\gamma_0/\gamma_1)$, similarly to in the proof of Lemma 2. For (i), the second term on the RHS of Theorem 5 can be bounded above by $\lambda$, while for (ii) it vanishes entirely. $\qquad\square$

In each case, we can plug the resulting bound into Theorem 1 to get a version of Theorem 3 for the PF-ODE setting with the given noising schedule. We define $K_t = \lambda(|\dot{\gamma}_t|/\gamma_t) + \min\{\lambda(|\dot{\beta}_t|/\beta_t), \lambda^{1/2}R(|\dot{\beta}_t|/\gamma_t)\}$ and let $\mathcal{V}$ be the set of functions $v : \mathbb{R}^d \times [0, 1] \to \mathbb{R}^d$ which are $K_t$-Lipschitz in $\mathbf{x}$ for all $t \in [0, 1]$ as before.

**Theorem 6.** *Suppose that Assumptions 1-3 and 4' hold and we are in either (i) the VP ODE or (ii) the VE ODE setting above. Then $v^X \in \mathcal{V}$ and for any $v_\theta \in \mathcal{V}$, if $Y$ is a flow starting in $\tilde{\pi}_0$ with velocity field $v_\theta$ and $\hat{\pi}_1$ is the law of $Y_1$, then*

> *(i) in the VP ODE setting, we have $W_2(\hat{\pi}_1, \tilde{\pi}_1) \leq \varepsilon(e/\gamma_1)^\lambda$;*

> *(ii) in the VE ODE setting, we have $W_2(\hat{\pi}_1, \tilde{\pi}_1) \leq \varepsilon(1/\gamma_1)^\lambda$.*

*Proof.* That $v^X \in \mathcal{V}$ follows analogously to the proof for Theorem 3. The results then follow from combining Theorem 1 with the bounds in Corollary 2, as in the proof of Theorem 3. $\qquad\square$

## 5 Conclusion

We have provided the first bounds on the error of the general flow matching procedure that apply in the case of completely deterministic sampling. Under the smoothness criterion of Assumption 4 on the data distributions, we have derived bounds which for a given sufficiently smooth choice of $\alpha_t$, $\beta_t$, $\gamma_t$ and fixed data distribution are polynomial in the level of Gaussian smoothing $\gamma_{\max}/\gamma_{\min}$ and the $L^2$ error $\varepsilon$ of our velocity approximation.

However, our bounds still depend exponentially on the parameter $\lambda$ from Assumption 4. It is therefore a key question how $\lambda$ behaves for typical distributions or as we scale up the dimension. In particular, the informal argument we provide in Section A.2 suggests that $\lambda$ may scale linearly with $d$. However, we expect that in many practical cases $\lambda$ should be $\Theta(1)$ even as $d$ scales. Furthermore, even if this is not the case, in the proof of Theorem 1 we only require control of the norm of $\nabla_{\mathbf{x}} v_\theta(\mathbf{x}, t)$ when applied to $\nabla_{\mathbf{x}} Y_{s,t}^{\mathbf{x}}$. In cases such as these, the operator norm bound is typically loose unless $\nabla_{\mathbf{x}} Y_{s,t}^{\mathbf{x}}$ is highly correlated with the largest eigenvectors of $\nabla_{\mathbf{x}} v_\theta(\mathbf{x}, t)$. We see no a priori reason why these two should be highly correlated in practice, and so we expect in most practical applications to get behaviour which is much better than linear in $d$. Quantifying this behaviour remains a question of interest.

Finally, the fact that we get weaker results in the fully deterministic setting than Albergo et al. (2023) do when adding a small amount of Gaussian noise in the reverse sampling procedure suggests that some level of Gaussian smoothing is helpful for sampling and leads to the suppression of the exploding divergences of paths.

## Acknowledgements

We thank Michael Albergo, Nicholas Boffi and Eric Vanden-Eijnden for their comments on an early version of this paper. Joe Benton was supported by the EPSRC through the StatML CDT (EP/S023151/1). Arnaud Doucet acknowledges support from EPSRC grants EP/R034710/1 and EP/R018561/1.

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

# A    Exploration of Definition 1

## A.1    Special cases of $\lambda$-regularity

First, we show that all log-concave random variables are $\lambda$-regular for $\lambda = 1$. The key ingredient in the proof is the following result of Brascamp & Lieb (1976).

**Proposition 3** (Brascamp-Lieb 1976). *Suppose that $W$ is an $\mathbb{R}^d$-valued random variable with density function $p_W(\mathbf{x}) = e^{-\varphi(\mathbf{x})}$, where $\varphi$ is strictly convex on $\mathbb{R}^d$ and twice differentiable. Assume that $D^2\varphi \geq \mu I_d$ with $\mu > 0$ and $g \in C^1(\mathbb{R}^d)$. Then,*

$$\mathrm{Var}_W(g(W)) \leq \mathbb{E}\left[\langle (D^2\varphi)^{-1}\nabla g(W), \nabla g(W)\rangle\right] \leq \frac{1}{\mu}\mathbb{E}\left[\|\nabla g(W)\|^2\right].$$

**Lemma 4.** *Suppose that $W$ is an $\mathbb{R}^d$-valued log-concave random variable. Then $W$ is $\lambda$-regular for $\lambda = 1$.*

*Proof.* Fix $\tau > 0$ and denote the density of $W$ by $p_W(\mathbf{x}) = e^{-\varphi(\mathbf{x})}$ for some convex function $\varphi$. Take $\xi \sim \mathcal{N}(0, \tau^2 I_d)$ and set $W' = W + \xi$. Then, the density of $\xi$ conditional on $W'$ is given by

$$p_{\xi|W'}(\xi|\mathbf{x}') \propto e^{-\|\xi\|^2/(2\tau^2)}p_W(\mathbf{x}' - \xi) = \exp\left\{-\frac{\|\xi\|^2}{2\tau^2} - \varphi(\mathbf{x}' - \xi)\right\}.$$

Let $\mathbf{u} \in \mathbb{R}^d$ be an arbitrary unit vector, and consider $g(W) = \mathbf{u}^T W$. Since $\varphi'(\xi) = \frac{\|\xi\|^2}{2\tau^2} + \varphi(\mathbf{x}' - \xi)$ is strictly convex in $\xi$, and

$$D^2\varphi' = \tau^{-2}I_d + D^2\varphi(\mathbf{x}' - \xi) \geq \tau^{-2}I_d,$$

we can apply Theorem 3 to the random variable $\xi$ conditional on $W'$ with $\mu = \tau^{-2}$ to get

$$\mathrm{Var}_{\xi|W'}(\mathbf{u}^T\xi) \leq \tau^2\mathbb{E}\left[\|\mathbf{u}\|_2^2\right] = \tau^2.$$

Then,

$$\left\|\mathrm{Cov}_{\xi|W'}(\xi)\right\|_{\mathrm{op}} \leq \sup_{\|\mathbf{u}\|_2=1}\mathrm{Var}_{\xi|W'}(\mathbf{u}^T\xi) \leq \tau^2.$$

$\square$

Second, we show all random variables which are Gaussian on a linear subspace of $\mathbb{R}^d$ are $\lambda$-regular for $\lambda = 1$.

**Lemma 5.** *Suppose that $W$ is an $\mathbb{R}^d$-valued random variable supported on some linear subspace $\mathcal{S} \subseteq \mathbb{R}^d$ and that $W$ restricted to $\mathcal{S}$ is Gaussian with positive definite covariance matrix. Then $W$ is $\lambda$-regular for $\lambda = 1$.*

*Proof.* We decompose $\xi$ into two orthogonal components $\xi_\perp$ and $\xi_\parallel$, so that $\xi = \xi_\perp + \xi_\parallel$, and $\xi_\perp$ is perpendicular to $\mathcal{S}$ while $\xi_\parallel$ is parallel to $\mathcal{S}$. Then, we can write $W' = (W + \xi_\parallel) + \xi_\perp$. We denote $W'_\parallel = W + \xi_\parallel$ and note that $W'_\parallel \in \mathcal{S}$. From observing $W' = \mathbf{x}$ we can deduce the values of both $W'_\parallel$ and $\xi_\perp$. Therefore, $\mathrm{Cov}_{\xi|W'=\mathbf{x}}(\xi) = \mathrm{Cov}_{\xi|W'=\mathbf{x}}(\xi_\parallel) = \mathrm{Cov}_{\xi_\parallel|W'_\parallel=\mathbf{x}_\parallel}(\xi_\parallel)$, where $\mathbf{x}_\parallel$ denotes the projection of $\mathbf{x}$ onto $\mathcal{S}$.

By restricting our attention to the subspace $\mathcal{S}$, we may therefore reduce to the case where $W$ is Gaussian with full support. The result then follows from Lemma 4, since Gaussians with full support are log-concave.    $\square$

Third, we show that all random variables which are locally at least as smooth as a Gaussian of covariance $\sigma^2 I_d$ and are bounded up to Gaussian tails of covariance $\sigma^2 I_d$ are $\lambda$-regular.

**Lemma 6.** *Suppose that $W$ is an $\mathbb{R}^d$-valued random variable which can be decomposed as $W = U + \eta$, where $U$ and $\eta$ are independent random variables such that $\|U\| \leq R$ for some $R > 0$ and $\eta \sim \mathcal{N}(0, \sigma^2 I_d)$ for some $\sigma > 0$. Then $W$ is $\lambda$-regular for $\lambda = 1 + (R^2/\sigma^2)$.*

*Proof.* Fix $\tau > 0$, so that $W' = W + \xi = U + \eta + \xi$ where $\eta \sim \mathcal{N}(0, \sigma^2 I_d)$ and $\xi \sim \mathcal{N}(0, \tau^2 I_d)$ and $U, \eta, \xi$ are all independent. By the law of total variance, we have

$$\mathrm{Cov}_{\xi|W'}(\xi) = \mathbb{E}\left[\mathrm{Cov}_{\xi|W',U}(\xi) \mid W'\right] + \mathrm{Cov}(\mathbb{E}\left[\xi \mid W', U\right] \mid W').$$

The distribution of $\xi \mid W', U$ is the same as the distribution of $\xi \mid (\eta + \xi)$, so it suffices to understand the latter. To this end, we define $\rho = \eta + \xi$ and $\omega = \sigma^2 \xi - \tau^2 \eta$. It is straightforward to check that $\rho$ and $\omega$ are independent centered Gaussians with covariances $(\sigma^2 + \tau^2)I_d$ and $\sigma^2 \tau^2 (\sigma^2 + \tau^2)I_d$ respectively. Then, we can write $\xi = \frac{1}{\sigma^2 + \tau^2}(\tau^2 \rho + \omega)$, from which it follows that

$$\text{Cov}_{\xi \mid W', U}(\xi) = \left(\tfrac{1}{\sigma^2 + \tau^2}\right)^2 \text{Cov}(\tau^2 \rho + \omega \mid \rho) = \left(\tfrac{\sigma^2 \tau^2}{\sigma^2 + \tau^2}\right) I_d.$$

Therefore, $\mathbb{E}\left[\text{Cov}_{\xi \mid W', U}(\xi) \mid W'\right] = \left(\tfrac{\sigma^2 \tau^2}{\sigma^2 + \tau^2}\right) I_d$. In addition, we have

$$\mathbb{E}\left[\xi \mid W', U\right] = \left(\tfrac{1}{\sigma^2 + \tau^2}\right) \mathbb{E}\left[\tau^2 \rho + \omega \mid \rho\right] = \left(\tfrac{\tau^2}{\sigma^2 + \tau^2}\right) \rho,$$

so $\text{Cov}(\mathbb{E}\left[\xi \mid W', U\right] \mid W') = \left(\tfrac{\tau^2}{\sigma^2 + \tau^2}\right)^2 \text{Cov}_{\eta, \xi \mid W'}(\eta + \xi)$. Finally, we have

$$\left\|\text{Cov}_{\eta, \xi \mid W'}(\eta + \xi)\right\|_{\text{op}} = \left\|\text{Cov}_{U \mid W'}(W' - U)\right\|_{\text{op}} = \left\|\text{Cov}_{U \mid W'}(U)\right\|_{\text{op}} \leq R^2.$$

Putting this all together, we see that

$$\left\|\text{Cov}_{\xi \mid W'}(\xi)\right\|_{\text{op}} \leq \left\|\mathbb{E}\left[\text{Cov}_{\xi \mid W', U}(\xi) \mid W'\right]\right\|_{\text{op}} + \left\|\text{Cov}(\mathbb{E}\left[\xi \mid W', U\right] \mid W')\right\|_{\text{op}}$$

$$\leq \frac{\sigma^2 \tau^2}{\sigma^2 + \tau^2} + R^2 \left(\frac{\tau^2}{\sigma^2 + \tau^2}\right)^2$$

$$\leq \lambda \tau^2$$

for $\lambda = 1 + (R^2/\sigma^2)$, as required. $\qquad\square$

We note in particular that Lemma 6 can be applied in the case where our target distribution is a bounded mixture of Gaussian components with covariance $\sigma^2 I_d$ for some $\sigma > 0$. We obtain the following corollary.

**Corollary 3.** *Suppose that* $\pi = \sum_{i=1}^K \mu_i \mathcal{N}(\mathbf{x}_i, \sigma^2 I_d)$ *is a mixture of Gaussian components of covariance* $\sigma^2 I_d$ *for some* $\sigma > 0$, *where the weights* $\mu_i$ *satisfy* $\sum_{i=1}^K \mu_i = 1$ *and we have* $\|\mathbf{x}_i\| \leq R$ *for all* $i = 1, \ldots, K$. *Then* $\pi$ *is* $\lambda$-*regular for* $\lambda = 1 + (R^2/\sigma^2)$.

*Proof.* This follows immediately from the fact that $\pi$ can be represented in the form required to apply Lemma 6, by taking $U$ to have distribution $\sum_{i=1}^K \mu_i \delta_{\mathbf{x}_i}$. $\qquad\square$

## A.2 High-probability bounds

**Lemma 7.** *Suppose that* $W$ *is an* $\mathbb{R}^d$-*valued random variable. For any* $\tau > 0$, *if we take* $\xi \sim \mathcal{N}(0, \tau^2 I_d)$ *and set* $W' = W + \xi$, *then we have*

$$\left\|\text{Cov}_{\xi \mid W'}(\xi)\right\|_{\text{op}} \leq 2dc^2 \tau^2$$

*with probability at least* $1 - 6de^{-c^2/2}$ *for any* $c \geq 1$.

*Proof.* First, we have

$$\left\|\text{Cov}_{\xi \mid W'}(\xi)\right\|_{\text{op}} \leq \sup_{\|\mathbf{u}\|_2 = 1} \text{Var}_{\xi \mid W'}(\mathbf{u}^T \xi) \leq \sup_{\|\mathbf{u}\|_2 = 1} \mathbb{E}_{\xi \mid W'}\left[(\mathbf{u}^T \xi)^2\right] \leq \mathbb{E}\left[\|\xi\|_2^2 \mid W'\right].$$

Next, we bound the quantity $\mathbb{E}\left[\|\xi\|_2^2 \mid W'\right]$ using Markov's inequality. If $\|\xi\|_2^2 > dc^2 \tau^2$, then we must have $\xi_i^2 > c^2 \tau^2$ for some $i$ between 1 and $d$. Therefore,

$$\mathbb{E}\left[\|\xi\|_2^2 \mathbb{1}_{\|\xi\|_2^2 > dc^2 \tau^2}\right] \leq d\,\mathbb{E}\left[\|\xi\|_2^2 \mathbb{1}_{\xi_1^2 > c^2 \tau^2}\right]$$

$$= d\,\mathbb{E}\left[\xi_1^2 \mathbb{1}_{\xi_1^2 > c^2 \tau^2}\right] + d\sum_{i=2}^d \mathbb{E}\left[\xi_i^2 \mathbb{1}_{\xi_1^2 > c^2 \tau^2}\right]$$

$$= d\,\mathbb{E}\left[\xi_1^2 \mathbb{1}_{\xi_1^2 > c^2 \tau^2}\right] + d(d-1)\tau^2 \mathbb{P}\left(\xi_1^2 > c^2 \tau^2\right).$$

A standard Chernoff bound gives $\mathbb{P}\left(\xi_1^2 > c^2\tau^2\right) \leq 2e^{-c^2/2}$, and

$$
\begin{aligned}
\mathbb{E}\left[\xi_1^2 \mathbb{1}_{\xi_1^2 > c^2\tau^2}\right] &= 2\int_{c\tau}^{\infty} \frac{z^2}{\sqrt{2\pi\tau^2}} e^{-z^2/(2\tau^2)} \,\mathrm{d}z \\
&= 2\tau^2 \int_c^{\infty} \frac{z^2}{\sqrt{2\pi}} e^{-z^2/2} \,\mathrm{d}z \\
&= 2\tau^2 \left[-\frac{z}{\sqrt{2\pi}} e^{-z^2/2}\right]_c^{\infty} + 2\tau^2 \int_c^{\infty} \frac{1}{\sqrt{2\pi}} e^{-z^2/2} \,\mathrm{d}z \\
&= \frac{2\tau^2 c}{\sqrt{2\pi}} e^{-c^2/2} + 2\tau^2 \mathbb{P}\left(\xi_1 > c\tau\right) \\
&\leq 2\tau^2 e^{-c^2/2}\left(\frac{c}{\sqrt{2\pi}} + 1\right) \\
&\leq 4c\tau^2 e^{-c^2/2}.
\end{aligned}
$$

Hence

$$
\begin{aligned}
\mathbb{E}\left[\|\xi\|_2^2 \mathbb{1}_{\|\xi\|_2^2 > dc^2\tau^2}\right] &\leq 4dc\tau^2 e^{-c^2/2} + 2d^2\tau^2 e^{-c^2/2} \\
&\leq 6d^2 c\tau^2 e^{-c^2/2}.
\end{aligned}
$$

It follows by Markov's inequality that

$$
\begin{aligned}
\mathbb{P}\left(\mathbb{E}\left[\|\xi\|_2^2 \mathbb{1}_{\|\xi\|_2^2 > dc^2\tau^2} \mid W'\right] \geq dc^2\tau^2\right) &\leq \frac{\mathbb{E}\left[\|\xi\|_2^2 \mathbb{1}_{\|\xi\|_2^2 > dc^2\tau^2}\right]}{dc^2\tau^2} \\
&\leq 6de^{-c^2/2}.
\end{aligned}
$$

Finally, we can write

$$
\mathbb{E}\left[\|\xi\|_2^2 \mid W'\right] \leq \mathbb{E}\left[\|\xi\|_2^2 \mathbb{1}_{\|\xi\|_2^2 > dc^2\tau^2} \mid W'\right] + dc^2\tau^2,
$$

and so $\left\|\mathrm{Cov}_{\xi|W'}(\xi)\right\|_{\mathrm{op}} \leq \mathbb{E}\left[\|\xi\|_2^2 \mid W'\right] \leq 2dc^2\tau^2$ with probability at least $1 - 6de^{-c^2/2}$, as required. $\qquad\square$

# B   Derivation of formulae for gradients of velocity fields

## B.1   Proof of Lemma 1

In order to prove Lemma 1, we use the following intermediate result.

**Lemma 8.** *If $X$ is the stochastic interpolant between $\pi_0$ and $\pi_1$, then*

$$
\nabla_{\mathbf{x}} \mathbb{E}\left[X_0 \mid X_t = \mathbf{x}\right] = -\frac{1}{\gamma_t} \mathrm{Cov}_{\mathbf{x}}(X_0, Z).
$$

*Moreover, a similar expression holds for $X_1$ in place of $X_0$.*

*Proof.* First, note that $X_t | X_0, X_1 \sim \mathcal{N}(\alpha_t X_0 + \beta_t X_1, \gamma_t^2 I_d)$, so

$$
\nabla_{\mathbf{x}} \log(p_{X_t|X_0,X_1}(\mathbf{x}|\mathbf{x}_0,\mathbf{x}_1)) = -\frac{1}{\gamma_t^2}(\mathbf{x} - \alpha_t \mathbf{x}_0 - \beta_t \mathbf{x}_1)^T.
$$

Therefore,

$$
\begin{aligned}
\nabla_{\mathbf{x}} \log(p_{X_t}(\mathbf{x})) &= \frac{1}{p_{X_t}(\mathbf{x})} \cdot \nabla_{\mathbf{x}}\left(\int_{\mathbb{R}^d}\int_{\mathbb{R}^d} p_{X_t|X_0,X_1}(\mathbf{x}|\mathbf{x}_0,\mathbf{x}_1) p_{X_0,X_1}(\mathbf{x}_0,\mathbf{x}_1) \,\mathrm{d}\mathbf{x}_0\mathrm{d}\mathbf{x}_1\right) \\
&= \frac{1}{p_{X_t}(\mathbf{x})} \int_{\mathbb{R}^d}\int_{\mathbb{R}^d} p_{X_t|X_0,X_1}(\mathbf{x}|\mathbf{x}_0,\mathbf{x}_1) p_{X_0,X_1}(\mathbf{x}_0,\mathbf{x}_1) \nabla_{\mathbf{x}} \log(p_{X_t|X_0,X_1}(\mathbf{x}|\mathbf{x}_0,\mathbf{x}_1)) \,\mathrm{d}\mathbf{x}_0\mathrm{d}\mathbf{x}_1 \\
&= -\frac{1}{\gamma_t^2}\mathbb{E}\left[(X_t - \alpha_t X_0 - \beta_t X_1)^T \mid X_t = \mathbf{x}\right]
\end{aligned}
$$

We can then calculate

$$
\begin{aligned}
\nabla_{\mathbf{x}} \mathbb{E}\big[X_0 \mid X_t = \mathbf{x}\big] &= \nabla_{\mathbf{x}} \left( \frac{1}{p_{X_t}(\mathbf{x})} \int_{\mathbb{R}^d} \int_{\mathbb{R}^d} \mathbf{x}_0 \, p_{X_t|X_0,X_1}(\mathbf{x}|\mathbf{x}_0,\mathbf{x}_1) p_{X_0,X_1}(\mathbf{x}_0,\mathbf{x}_1) \, \mathrm{d}\mathbf{x}_0 \mathrm{d}\mathbf{x}_1 \right) \\
&= \frac{1}{p_{X_t}(\mathbf{x})} \int_{\mathbb{R}^d} \int_{\mathbb{R}^d} \mathbf{x}_0 \, p_{X_t|X_0,X_1}(\mathbf{x}|\mathbf{x}_0,\mathbf{x}_1) p_{X_0,X_1}(\mathbf{x}_0,\mathbf{x}_1) \nabla_{\mathbf{x}} \big( \log p_{X_t|X_0,X_1}(\mathbf{x}|\mathbf{x}_0,\mathbf{x}_1) \big) \, \mathrm{d}\mathbf{x}_0 \mathrm{d}\mathbf{x}_1 \\
&\quad - \big( \nabla_{\mathbf{x}} \log p_{X_t}(\mathbf{x}) \big) \left( \frac{1}{p_{X_t}(\mathbf{x})} \int_{\mathbb{R}^d} \int_{\mathbb{R}^d} \mathbf{x}_0 \, p_{X_t|X_0,X_1}(\mathbf{x}|\mathbf{x}_0,\mathbf{x}_1) p_{X_0,X_1}(\mathbf{x}_0,\mathbf{x}_1) \, \mathrm{d}\mathbf{x}_0 \mathrm{d}\mathbf{x}_1 \right) \\
&= -\frac{1}{\gamma_t^2} \mathbb{E}_{\mathbf{x}}\big[ X_0(X_t - \alpha_t X_0 - \beta_t X_1)^T \big] + \frac{1}{\gamma_t^2} \mathbb{E}_{\mathbf{x}}\big[ (X_t - \alpha_t X_0 - \beta_t X_1)^T \big] \mathbb{E}_{\mathbf{x}}\big[ X_0 \big] \\
&= -\frac{1}{\gamma_t} \mathrm{Cov}_{\mathbf{x}}(X_0, Z).
\end{aligned}
$$

$\square$

**Lemma 1.** *If $X$ is the stochastic interpolant between $\pi_0$ and $\pi_1$, then $v^X(\mathbf{x}, t)$ is differentiable with respect to $\mathbf{x}$ and*

$$
\nabla_{\mathbf{x}} v^X(\mathbf{x}, t) = \frac{\dot{\gamma}_t}{\gamma_t} I_d - \frac{1}{\gamma_t} \mathrm{Cov}_{\mathbf{x}}(\dot{X}_t, Z).
$$

*Proof.* Since $X_t = \alpha_t X_0 + \beta_t X_1 + \gamma_t Z$ and $\dot{X}_t = \dot{\alpha}_t X_0 + \dot{\beta}_t X_1 + \dot{\gamma}_t Z$, we can write

$$
\mathbb{E}\big[ \dot{X}_t \mid X_t = \mathbf{x}, X_0 = \mathbf{x}_0, X_1 = \mathbf{x}_1 \big] = \dot{\alpha}_t \mathbf{x}_0 + \dot{\beta}_t \mathbf{x}_1 + \frac{\dot{\gamma}_t}{\gamma_t} (\mathbf{x} - \alpha_t \mathbf{x}_0 - \beta_t \mathbf{x}_1)
$$

and therefore

$$
\mathbb{E}\big[ \dot{X}_t \mid X_t = \mathbf{x} \big] = \frac{\dot{\gamma}_t}{\gamma_t} \mathbf{x} + \frac{(\dot{\alpha}_t \gamma_t - \dot{\gamma}_t \alpha_t)}{\gamma_t} \cdot \mathbb{E}\big[ X_0 \mid X_t = \mathbf{x} \big] + \frac{(\dot{\beta}_t \gamma_t - \dot{\gamma}_t \beta_t)}{\gamma_t} \cdot \mathbb{E}\big[ X_1 \mid X_t = \mathbf{x} \big].
$$

Taking gradients with respect to $\mathbf{x}$ and applying Lemma 8,

$$
\begin{aligned}
\nabla_{\mathbf{x}} v^X(\mathbf{x}, t) &= \frac{\dot{\gamma}_t}{\gamma_t} I_d - \frac{(\dot{\alpha}_t \gamma_t - \dot{\gamma}_t \alpha_t)}{\gamma_t^2} \cdot \mathrm{Cov}_{\mathbf{x}}(X_0, Z) - \frac{(\dot{\beta}_t \gamma_t - \dot{\gamma}_t \beta_t)}{\gamma_t^2} \cdot \mathrm{Cov}_{\mathbf{x}}(X_1, Z) \\
&= \frac{\dot{\gamma}_t}{\gamma_t} I_d + \frac{\dot{\gamma}_t}{\gamma_t^2} \mathrm{Cov}_{\mathbf{x}}(X_t - \gamma_t Z, Z) - \frac{1}{\gamma_t} \mathrm{Cov}_{\mathbf{x}}(\dot{\alpha}_t X_0 + \dot{\beta}_t X_1, Z) \\
&= \frac{\dot{\gamma}_t}{\gamma_t} I_d - \frac{1}{\gamma_t} \mathrm{Cov}_{\mathbf{x}}(\dot{X}_t, Z).
\end{aligned}
$$

$\square$

## B.2 Proof of Lemma 3

**Lemma 3.** *If $X$ is the stochastic interpolant in the PF-ODE setting above, then $v^X(\mathbf{x}, t)$ is differentiable with respect to $\mathbf{x}$ and*

$$
\nabla_{\mathbf{x}} v^X(\mathbf{x}, t) = \frac{\dot{\gamma}_t}{\gamma_t} I_d - \left( \frac{\dot{\gamma}_t}{\gamma_t} - \frac{\dot{\beta}_t}{\beta_t} \right) \mathrm{Cov}_{\mathbf{x}}(Z).
$$

*Proof.* From Lemma 1, we have

$$
\nabla_{\mathbf{x}} v^X(\mathbf{x}, t) = \frac{\dot{\gamma}_t}{\gamma_t} I_d - \frac{1}{\gamma_t} \mathrm{Cov}_{\mathbf{x}}(\dot{X}_t, Z).
$$

Then, since $\alpha_t = 0$ we have $X_t = \beta_t X_1 + \gamma_t Z$ and $\dot{X}_t = \dot{\beta}_t X_1 + \dot{\gamma}_t Z$, so we can write

$$
\begin{aligned}
\nabla_{\mathbf{x}} v^X(\mathbf{x}, t) &= \frac{\dot{\gamma}_t}{\gamma_t} I_d - \frac{1}{\gamma_t} \mathrm{Cov}_{\mathbf{x}} \left( \frac{\dot{\beta}_t}{\beta_t} (X_t - \gamma_t Z) + \dot{\gamma}_t Z, Z \right) \\
&= \frac{\dot{\gamma}_t}{\gamma_t} I_d - \left( \frac{\dot{\gamma}_t}{\gamma_t} - \frac{\dot{\beta}_t}{\beta_t} \right) \mathrm{Cov}_{\mathbf{x}}(Z),
\end{aligned}
$$

noting that we may discard the $X_t$ term since we are conditioning on $X_t = \mathbf{x}$. $\square$

