# OpenReview forum: "Error Bounds for Flow Matching Methods"
_TMLR — Accepted by TMLR_

### Review · Reviewer_W9Ya · 2023-12-03

**Summary Of Contributions:**

The paper presents bounds on the error of flow matching for the case of fully deterministic sampling when the data distribution does not have full support. The first result in the paper establishes a bound on the error of the flow matching approximation under the 2-Wasserstein metric. The bound is expressed in terms of the L^2 training error and the Lipschitz constant of the approximate velocity field. The second main result first demonstrates that, provided a smoothness assumption is satisfied, the true velocity field is Lipschitz, and subsequently bounds the Lipschitz constant. The two results are then combined to construct a bound on the approximation error which depends polynomially on the L^2 training error.

**Audience:**

Yes

**Broader Impact Concerns:**

No concerns.

**Claims And Evidence:**

Yes

**Requested Changes:**

The following changes are not critical, but additional discussion would strengthen the paper and make it more readily accessible and usable by practitioners of flow matching who would like to tailor results to their particular problem setting.

(1) The key challenges in applying Theorem 3 appear to be identification of R, \lambda, and \epsilon. As a practitioner, if I were presented with a dataset that I would like to model, and I wanted to make reasonable assumptions about the target distribution generating the data, and I selected a particular neural network to learn the approximate flow, could I evaluate or bound these three quantities in any way? Is it possible to tie them back (even if it makes the bounds looser) to quantities that can be evaluated in a particular problem setting under reasonable assumptions about the distribution?

(2) Theorem 4 leaves work for the reader to do to plug the results back into Theorem 1 to construct a result like Theorem 3 that is more direct and easier to understand. It would be helpful to provide corollaries, even if only in the appendix, to specify the results that arise in these common settings.

**Strengths And Weaknesses:**

Strengths
(1)	The paper provides a valuable extension of bounds for flow matching, targeting the important practical setting of deterministic sampling and distributions without full support.
(2)	The paper is very well written, with good organization, and provides a clear exposition of the main results.
(3)	The assumptions are discussed and, for the most part, they are justified and the motivation for adopting them is clearly explained.
Weaknesses
(1)	Despite a good effort to make the assumptions reasonable and not particularly onerous, the paper doesn’t provide a clear example of how a practitioner might use the results. It would be helpful to understand, what steps a when presented with a dataset and making simple, reasonable assumptions about the approximation error and nature of the distribution of the data, to construct a bound. Perhaps the intention is to provide x
(2)	As the authors specify in the conclusion, the bounds depend exponentially on a smoothness parameter and it is not obvious from the provided results how this parameter scales as the dimension increases or how it behaves for commonly encountered data distributions.

Overall, this is a strong paper that provides a useful extension of existing bounds for flow matching. There are practical advantages to deterministic sampling cases and many distributions of interest do not have full support.

While I think the paper is publishable in its current form, since it contributes to the theoretical understanding of flow matching in a valuable way, I think it could benefit from a little extra effort to make it clearer how a practitioner might make use of these results (or whether they even can). It would be good to be identify which assumptions are unverifiable and which components of the bounds are unquantifiable.

---

> ### Author Response · Authors · 2023-12-31
> **Part 1**
>
> We would like to thank the reviewer for their kind remarks and thoughtful comments on our paper.
>
> > The key challenges in applying Theorem 3 appear to be identification of $R$, $\lambda$, and $\varepsilon$. As a practitioner, if I were presented with a dataset that I would like to model, and I wanted to make reasonable assumptions about the target distribution generating the data, and I selected a particular neural network to learn the approximate flow, could I evaluate or bound these three quantities in any way? Is it possible to tie them back (even if it makes the bounds looser) to quantities that can be evaluated in a particular problem setting under reasonable assumptions about the distribution?
>
> In general, unless we know something a priori about the data generating distribution, it will be difficult to verify the assumptions of our work in practice. In some situations (e.g. image inputs with bounded pixel values), we will have access to a bound on $R$. However, this will not always be the case, and estimating the radius of the support of $\pi_0, \pi_1$ may be hard in general. Similarly, estimating $\lambda$ is likely to be challenging in practice (though it is the case that for certain classes of distributions we have tractable bounds on $\lambda$ - see Appendix A.1 - and so if we know that our distribution comes from one of these classes then we will have access to such a bound). In general, while we cannot empirically verify our assumptions for many datasets, we hope that our assumptions are broad enough to capture a very wide class of potential distributions.
>
> Estimating $\varepsilon$ is also in general intractable. Indeed, we train using the equivalent objective (3) rather than performing gradient descent on $\cal{L}(v)$ directly for precisely this reason. Because we minimize a proxy loss and never have access to the true drift, our bounds will inevitably be unquantifiable. With that said, our Assumption 1 on the error of the flow matching procedure is standard in the literature (see e.g. Albergo and Vanden-Eijnden (2023) or Chen et al. (2023a;c)), and analysing its suitability was somewhat orthogonal to the main thesis of our work.
>
> > As the authors specify in the conclusion, the bounds depend exponentially on a smoothness parameter and it is not obvious from the provided results how this parameter scales as the dimension increases or how it behaves for commonly encountered data distributions.
>
> We have tried to provide some intuition for how the smoothness parameter $\lambda$ will scale with dimension at the end of Section 3.2 after Definition 1 -- it turns out that for any distribution, it is intuitively "$\lambda$-regular with high probability" for some $\lambda$ which scales linearly with $d$. However, making this into a formal statement that held everywhere (as is required by Definition 1) turned out to be surprisingly hard - we expect that formally controlling $\lambda$ as a function of $d$ likely requires additional assumptions on the data distribution.
>
> In the updated version of the manuscript, we have also studied how the smoothness parameter behaves for distributions which can be expressed as a mixture of Gaussians. Roughly, we find that for distributions which are bounded (except for Gaussian tails of covariance $\sigma^2 I_d$ and at least as locally smooth as a covariance $\sigma^2 I_d$ Gaussian, they are $\lambda$-regular for $\lambda = 1 + (R^2 / \sigma^2)$. For details of this new result, see our reply to Reviewer 28fW and Section 3.2 and Appendix A.1 of the updated manuscript. Importantly, this class of distributions can approximate a range of common distributions, including highly multimodal distributions.
>
> > Theorem 4 leaves work for the reader to do to plug the results back into Theorem 1 to construct a result like Theorem 3 that is more direct and easier to understand. It would be helpful to provide corollaries, even if only in the appendix, to specify the results that arise in these common settings.
>
> We have now moved the discussion following Theorem 4 into a pair of standalone results. Corollary 2 deals with the bounds on $\int_0^1 L^\ast_t \textup{d} t$ in the VP and VE ODE cases, while Theorem 6 explains how they can be used to construct an analogous result to Theorem 3 in the PF-ODE setting.

---

### Review · Reviewer_juW2 · 2023-12-09

**Summary Of Contributions:**

In this paper, the authors propose an error bound for flow matching bounds for determinist matching when the data distribution does not have full support. Real data in high dimensions does not have full support, so being able to prove these results is important to advance the theory behind flow models. From this perspective, the paper presents a significant contribution.

**Audience:**

Yes

**Claims And Evidence:**

Yes

**Requested Changes:**

The main changes are about a discussion on Theorem 3 limitations when we add the relaxations in Section 3.2

**Strengths And Weaknesses:**

My concern (and question to the authors) to help me assess the final decision about this paper is about the main result of the paper in Theorem 3. The bound depends on $\gamma_{max}/\gamma_{min}$ and $\gamma_t\in[0,\infty)$. This ratio can be very large (even infinity if $\gamma_{min}$ is 0). I am especially concerned with $\gamma_{min}$ being zero or close to zero. The authors relax the problem to avoid $\gamma_{min}$ being zero for $\gamma_0$ or $\gamma_1$ in Section 3.2 (second paragraph), but this does not ensure that $\gamma_t$ can be very small or close to zero in the path from $t=0$ to $t=1$. Also, we need $\gamma_0$ to be small to have a good approximation between $\pi_0$ and $\tilde{\pi}_0$ (same of $\gamma_1$ and $\pi_1$), but at the same time, the better this approximation is the larger the bound in Theorem 3 is. It is unclear how this trade-off can be understood in the paper. In a way, we have an uncertainty principle at play here. If we make $\tilde{\pi}_1$ be close to $\pi_1$ the bound on the error will be very large. We can only have a small bound if $\tilde{\pi}_1$ is not that close to $\pi_1$, but in that case, are we drawing samples from $\pi_1$? It will be good to understand this trade-off in the paper.

Second question, if we relax $\gamma_0$ and $\gamma_1$ away from zero and $\alpha_0$ and $\beta_1$ away from one, then $\tilde{\pi}_1$ would have full support because $\pi_0$ has full support, which would void one of the main claims of the paper. Is this the case, and if not why?

Minor comments:

In the second paragraph on Page 3, the authors say that the interpolation function is $I(x_0,x_1,t) = \alpha_t x_0+\beta_t x_1$ isn’t this linear interpolation too limited? Please comment.

In the second paragraph in Section 3.2 (page 6), the authors say that if it is simple to sample from $\pi_0$ then it is easy to sample from $\tilde{\pi}_0$. This might be trivial, but it is not to me. Can you tell me why is this the case?

---

> ### Author Response · Authors · 2023-12-31
> **Part 1**
>
> We thank the reviewer for their comments on our work.
>
> > I am especially concerned with $\gamma_{\min}$ being zero or close to zero. The authors relax the problem to avoid $\gamma_{\min}$ being zero for $\gamma_0$ or $\gamma_1$ in Section 3.2 (second paragraph), but this does not ensure that $\gamma_t$ can be very small or close to zero in the path from $t=0$ to $t=1$.
>
> In practice, the choice of the noise level $\gamma_t$ on the interpolating path is left up to the practitioner implementing the algorithm -- so, based on the results of Section 3, we see that we should pick the noise level such $\gamma_{\min}$ is at most polynomially small and $\gamma_t$ is far from zero on $(0,1)$ (as the reviewer remarks). Indeed, as we explain at the end of Section 3.2, this is possible in practice and leads to reasonable bounds. (Of course, as the reviewer notes, if we were to let $\gamma_{\min} = 0$ at any point on the path $[0,1]$ then our bounds would indeed become degenerate -- we intend to point out that this choice is not sensible in practice for precisely this reason, and can be avoided while only incurring a small cost in the total $W_2$ error of the approximation.)
>
> > Also, we need $\gamma_0$ to be small to have a good approximation between $\pi_0$ and $\tilde \pi_0$ (same of $\gamma_1$ and $\pi_1$), but at the same time, the better this approximation is the larger the bound in Theorem 3 is. It is unclear how this trade-off can be understood in the paper.
>
> The reviewer is correct to remark that there is a trade off here. Indeed, this is the trade off we seek to address in our remarks at the end of Section 3.3 -- which we have pulled out in Corollary 1 in the updated version of the manuscript for clarity. Here, we decompose the total $W_2$ error into two parts, $W_2(\hat \pi_1, \tilde \pi_1)$ which is decreasing in $\gamma_{\min}$, and $W_2(\tilde \pi_1, \pi_1)$ which  is increasing in $\gamma_{\min}$. In that section, we note that we may optimize the total bound over $\gamma_{\min}$, and observe that for a choice of $\gamma_{\min}$ on the order of $d^{-1/(4\lambda+2)} \varepsilon^{1 / (2 \lambda + 1)}$ the total bound becomes of order $d^{2 \lambda / (4 \lambda + 2)}\varepsilon^{1/(2 \lambda + 1)}$, which can be made arbitrarily small for sufficiently small $\delta$ and $\varepsilon$. This shows that there is a suitable choice of $\gamma_{\min}$ that successfully balances the trade off raised by the reviewer.
>
> > Second question, if we relax $\gamma_0$ and $\gamma_1$ away from zero and $\alpha_0$ and $\beta_1$ away from one, then $\tilde \pi_1$ would have full support because $\pi_0$ has full support, which would void one of the main claims of the paper. Is this the case, and if not why?
>
> It is indeed the case that once we add a small amount of Gaussian smoothing to $\pi_1$ to transform it into $\tilde \pi_1$ then it will have full support. However, it will also still be close in $W_2$ distance to the original distribution $\pi_1$. Therefore, by showing that the flow matching scheme provides a good approximation to $\tilde \pi_1$, we indirectly show that it also provides a good approximation to $\pi_1$. In this way, we are able to get good bounds on the approximation error for $\pi_1$, even when $\pi_1$ does not have full support, validating our claim that we obtain bounds for flow matching with fully deterministic sampling for distributions without full support.
>
> The only other work operating in the fully deterministic regime is Albergo \& Vanden-Eijnden (2023). But, their work cannot be applied directly to distributions without full support since they do not work with a time-varying Lipschitz constant -- this means that even after applying the above Gaussian smoothing method to reduce to distributions with full support, they would end up with a Lipschitz constant that becomes very large as $t \rightarrow 0,1$, making their bounds degenerate.
>
> > In the second paragraph on Page 3, the authors say that the interpolation function is $I(x_0, x_1, t) = \alpha_t x_0 + \beta_t x_1$ isn’t this linear interpolation too limited? Please comment.
>
> In fact, the range of interpolating paths that are captured by even this relatively simple linear form is very broad. For example, it includes all instances of the probability flow framework for diffusion models (Song et al., 2021), as well as the field that corresponds to an Optimal Transport displacement map between two Gaussian distributions (Lipman et al., 2023), and many others besides (Liu et al., 2022). Assuming a linear form of the interpolant function allows for the use of efficient procedures for sampling the forward process and learning the velocity flow (Lipman et al., 2023). As such, all practical implementations of flow matching of which we are aware employ a linear interpolant function, and so we chose to focus on linear interpolants in our work.

---

> ### Author Response · Authors · 2023-12-31
> **Part 2**
>
> > In the second paragraph in Section 3.2 (page 6), the authors say that if it is simple to sample from $pi_0$ then it is easy to sample from $\tilde \pi_0$. This might be trivial, but it is not to me. Can you tell me why is this the case?
>
> To sample from $\tilde \pi_0$, we may draw $X_0 \sim \pi_0$ and $Z \sim \mathcal{N}(0, I_d)$ independently, noting that these distributions are both easy to sample from, and set $\tilde X_0 = \alpha_0 X_0 + \gamma_0 Z$, possible since we know $\alpha_t$ and $\gamma_t$. As noted in Section 3.2, $\tilde X_0$ then has distribution $\tilde \pi_0$. We have added this explanation into the updated manuscript.

---

### Review · Reviewer_28fW · 2023-12-18

**Summary Of Contributions:**

This paper considers flow matching methods for sampling. Formally, to sample from a target distribution $\pi_1$, one can try to learn a deterministic flow from a distribution $\pi_0$, which is easy to sample from, to $\pi_1$. This flow can be specified by giving a path (together with a time-dependent speed) between any pair of points in space and, optionally, a time-dependent rate of injecting Gaussian noise. That is, to sample from the marginal distribution of the flow at time $t$, one samples $x_0, x_1$ from $\pi_0,\pi_1$ and forms $I_t(x_0,x_1) + \gamma_t \cdot g$ for $g\sim\mathcal{N}(0,I)$. This work considers linear paths, for which it is known that such flows can also be realized by ODEs, of which one example is the probability flow ODE from diffusion generative modeling.

The paper begins by giving a bounds on how robust the distribution at the end of the flow is to perturbations to the velocity field of the ODE. Under the assumption that the average squared perturbation along the path is bounded by $\epsilon^2$ and that the velocity fields are $L_t$-Lipschitz in space at time $t$, they show that the distribution changes in $W_2$ by at most $\epsilon\cdot \mathrm{exp}(\int^1_0 L_t \, \mathrm{d}t)$. This follows by a standard application of the Alekseev-Grobner formula.

At first blush, the exponential dependence on the integrated Lipschitzness might appear rather prohibitive; indeed, as the authors note, in the worst case the integrated Lipschitzness could be as large as the ambient dimension. The bulk of the paper is thus focused on showing that if the marginals of the process without Gaussian smoothing are sufficiently "regular", then the smoothed process' integrated Lipschitzness is actually not too large. For instance, if $\gamma_t$ is concave and the flow is "regular" in the sense that the marginal covariances are not too affected by Gaussian tilts , then $\int^1_0 L_t \,\mathrm{d}t$ can be bounded by $O(\log(\gamma_{\max}/\gamma_{\min}))$, where $\gamma_\max = \max_t \gamma_t$ and $\gamma_\min = \min_t \gamma_t$. The authors show that regularity holds under log-concavity.

**Audience:**

Yes

**Broader Impact Concerns:**

There are no adverse ethical implications of this work. Of course, with generative modeling there are the usual concerns about privacy, amplification of biases, etc., but this is well out of the scope of the submission.

**Claims And Evidence:**

Yes

**Requested Changes:**

I'm in favor of acceptance as I think it's a clean and self-contained result that should be of decent interest to TMLR's audiences. That said, implementing the following changes would strengthen the work in my view:

- Some of the technical discussion could benefit from being converted into formal lemmas. For instance, the $\log(\gamma_\max/\gamma_\min)$ dependence discussed at the end of P. 7 is invoked in the proof of Theorem 3 and should probably be moved into a standalone lemma statement.
- There is some discussion about how Appendix A.2 suggests regularity can scale with the dimension, but my understanding is that all that is done in Appendix A.2 is to show a fairly loose upper bound in terms of $d$. It would be more satisfying if the authors could provide a simple example of a distribution for which there is a matching lower bound
- The terminology "regularity" is a little overloaded, so it would be helpful for the authors to point out any prior works where the notion in Definition 1 is used. If not, perhaps they should note that this notion of regularity is non-standard and is the natural one that emerges from their proofs?

**Minor typos:**
- In Assumption 2, is Z^x_{s,s} supposed to start at x as well?
- Middle of P. 4: "smoothness assumption on v_{theta}(x,t) choose to make"
- Proposition 2: in the last equation in Proposition 2, should s in Y_s in the gradient be r?

**Strengths And Weaknesses:**

**Strengths**:
- There is significant interest, e.g. among people in the target audience for TMLR, in developing rigorous performance guarantees for diffusion models and related generative modeling frameworks. The present submission provides fairly clean and interpretable bounds for a broad class of generative models based on "flow-matching"
- The integrated Lipschitz constant is a natural quantity that has also arisen previously in discretization analyses for diffusion models, e.g. the ICML '23 paper of Chen-Daras-Dimakis on DDIMs, so it is nice that this paper initiates a principled study of this particular quantity and identifies some simple conditions under which it scales benignly.

**Weaknesses**:
- One somewhat unsatisfying aspect of the results is that it is unclear to what extent any of these bounds are tight. For instance, while the dependence on $d$ in the $W_2$ bound is polynomial, is the $\epsilon^{\Theta(1/\lambda)}$ dependence at the end of Section 3 necessary or just an artifact of the proof? The Conclusion touches upon the question of characterizing $\lambda$ for typical distributions and whether or not we should expect it to scale with the dimension, but there is the separate question of whether the exponential dependence on $1/\lambda$ is "real" to begin with.
- Relatedly, the paper only establishes regularity for fairly nice distributions like log-concave measures. One of the primary benefits of methods like sampling using the probability flow ODE is that one can hope to sample even from highly multi-modal distributions. How badly can regularity fail for such distributions? E.g. it would be nice to include bounds on $\lambda$ when the distribution is a mixture of separated Gaussians.

---

> ### Author Response · Authors · 2023-12-31
> **Part 1**
>
> Thank you for your thoughtful comments and engagement with the paper. In response to some of the questions you raise, we have the following remarks.
>
> > One somewhat unsatisfying aspect of the results is that it is unclear to what extent any of these bounds are tight. For instance, while the dependence on $d$ in the $W_2$ bound is polynomial, it is unclear to what extent any of these bounds are tight, is the $\varepsilon^{\Theta(1/\lambda)}$ dependence at the end of Section 3 necessary or just an artifact of the proof?
>
> The reviewer is correct to note that our work does not address the tightness of the bounds that we obtain. In general, we do not expect that our bounds are the tightest possible. Given that several works have shown better convergence under stronger smoothness results (see e.g. Li et al. (2023); Albergo et al. (2023)), it would not surprise us if the exponential dependence on $\varepsilon^{\Theta(1/\lambda)}$ was just an artifact of our proof.
>
> Nevertheless, we expect that some form of exponential dependence is likely inevitable if we want our results to hold for a sufficiently broad class of target distributions, because the difference between the flow ODEs will naturally explode exponentially unless we have some form of smoothing or contractivity assumption (as discussed briefly at the end of the introduction to Section 3). In particular, we expect this to be the case for distributions supported on a submanifold, such as atomic measures. We view characterising the largest natural class of distributions for which the exponential dependence can be avoided as an exciting avenue for future work.
>
> > Relatedly, the paper only establishes regularity for fairly nice distributions like log-concave measures. One of the primary benefits of methods like sampling using the probability flow ODE is that one can hope to sample even from highly multi-modal distributions. How badly can regularity fail for such distributions? E.g. it would be nice to include bounds on $\lambda$ when the distribution is a mixture of separated Gaussians.
>
> In fact, we can show that for mixtures of separated Gaussians of the form $\pi = \sum_{i=1}^K \mu_i \mathcal{N}(x_i, \sigma^2 I_d)$, where the weights $\mu_i$ satisfy $\sum_{i=1}^K \mu_i = 1$ and $\|x_i\| \leq R$ for $i = 1, \dots, K$, the distribution $\pi$ is $\lambda$-regular for $\lambda = 1 + (R^2 / \sigma^2)$. This shows that $\lambda$-regularity can hold even for highly multimodal distributions, so long as they are bounded (modulo Gaussian tails of covariance $\sigma^2 I_d$) and locally at least as smooth as a Gaussian distribution of covariance $\sigma^2 I_d$ for some $\sigma > 0$. We agree that the applicability of $\lambda$-regularity in the multimodal setting is of great practical relevance, and we have therefore included the above result in the updated manuscript in Appendix A.1, as well as a discussion of the result and its relevance for the multimodal setting in Section 3.2.
>
> Additionally, $\lambda$-regularity should hold for some value of $\lambda$ for arbitrary mixtures of separated Gaussians (or indeed for any distribution with moderately well-behaved tails). However, in the general Gaussian mixture setting $\lambda$ may depend very poorly on the parameters of the distribution (i.e. the covariances of the mixtures) -- since our final bounds depend exponentially on $\lambda$, we consider these to be practically equivalent to a failure of the $\lambda$-regularity condition and do not make a detailed attempt to quantify the dependence of $\lambda$ on the distribution parameters in these cases.
>
> > Some of the technical discussion could benefit from being converted into formal lemmas. For instance, the $\log (\gamma_{\max} / \gamma_{\min})$ dependence discussed at the end of P. 7 is invoked in the proof of Theorem 3 and should probably be moved into a standalone lemma statement.
>
> We have moved several pieces of technical discussion into standalone results in the updated version of the manuscript. The $\log (\gamma_{\max} / \gamma_{\min})$ dependence has been moved to Lemma 2, the discussion of the combined bounds at the end of Section 3.2 has been moved into Theorem 4 and Corollary 1, and the discussions of the VP and VE ODE applications have been moved into Corollary 2 and Theorem 6.

---

> ### Author Response · Authors · 2023-12-31
> **Part 2**
>
> > The terminology "regularity" is a little overloaded, so it would be helpful for the authors to point out any prior works where the notion in Definition 1 is used. If not, perhaps they should note that this notion of regularity is non-standard and is the natural one that emerges from their proofs?
>
> We agree that "regularity" is a bit overloaded in this literature. We are not aware of other works that use the version of $\lambda$-regularity that we use in this work (though the quantity controlled in Definition 1 bears some similarities to quantities arising in stochastic localization, which we mention briefly in Section 3.2). We have added a remark to the updated version of the manuscript explaining that our notion of regularity is non-standard and emerges from our proofs.
>
> In addition, thank you for pointing out the typos -- we have fixed these in the updated version of our manuscript.

---

### Author Response · Authors · 2023-12-31
**Updated version of manuscript**

We would like to thank all of the reviewers for their thoughtful comments on our work. As well as responses to individual questions from the reviewers, we have also provided an updated version of our manuscript. The main changes are in Section 3 and Appendix A.1, where we have:

- Provided a couple of additional results in Appendix A.1 showing that some Gaussian mixtures also satisfy $\lambda$-regularity, and explained the relevance of this result for understanding when $\lambda$-regularity might hold for highly multimodal distributions in Section 3.2.
- Extracted the discussion of the $\log(\gamma_{\max} / \gamma_{\min})$ dependence at the end of Section 3.2 into a standalone result (Lemma 2).
- Extracted the discussion at the end of Section 3.3 on how to combine the results of Section 3 into an end-to-end bound on the total Wasserstein error of the flow matching procedure into two results, Theorem 4 and Corollary 1.
- Reformatted the application of Theorem 5 to the specific cases of the VP and VE ODEs in Section 4 into two key results, Corollary 2 and Theorem 6.

Hopefully these changes have increased the clarity of our manuscript.

---

### Decision · Action_Editor_W71i · 2024-02-03

**Recommendation:** Accept as is

**Comment:**

The reviewers raised a few technical points and questioned some sharpness and reasonableness of assumptions (or ability to verify assumptions).  The authors' rebuttal addressed these issues, sometimes resolving them and other times merely acknowledging them as difficult issues. Overall, none of the reviewers nor myself think that these issues are significant enough to recommend against publication of the paper, and we were happy to see progress made on an exciting topic.

For more details, below is an abridged version of reviewer W9Ya's final summary:

**Strengths**
1. It provides interpretable bounds for flow matching, extending existing results to address the fully deterministic setting, allowing for distributions that do not have full support and relaxing regularity assumptions.
2. The results should be of interest to researchers exploring diffusion models - there is a strong desire for theoretical performance guarantees for these methods. While the results provided in this paper have aspects that are a little undesirable, they do constitute a useful stepping stone in the search for a more complete characterization.
3. The paper provides some useful discussion and results concerning regularity conditions and establishes that (mixture of Gaussian) multi-modal distributions satisfy the regularity required for the main results.

**Weaknesses mentioned by more than one reviewer (mostly addressed)**
1. The bounds are not particularly tight. In particular, there is an exponential dependence and it is not clear if this is necessary or an artifact of the proof technique. [Response: The authors conjecture that the exponential dependences is probably inevitable if the results target a broad class of distributions, and give a convincing informal argument]

2. The bounds are unquantifiable. For a given data distribution, one cannot establish whether the assumptions hold, nor evaluate or determine suitable constants. [Response: The authors have acknowledged that this is a limitation of the work. They point out that for some classes of distribution, the paper does derive some tractable bounds. Despite the limitation, the results still offer considerable value in terms of highlighting how the performance guarantees depend on aspects of the data distribution and the flow. The demonstration that a multi-modal distribution meets the required regularity condition is a valuable step in convincing a reviewer that the imposed assumptions are not unreasonably restrictive.]

**Audience:**

The paper is technical and the immediate audience is somewhat narrow, but this is an important new avenue of research, and new technical results from the literature, taken together, may lead to new insights in diffusion models (which are extremely popular these days, e.g., Stable Diffusion and Dall-E). I find that there is sufficient audience. Furthermore, while it is theoretical, given the closeness to applications, I think a machine learning venue like TMLR is appropriate.

**Claims And Evidence:**

The paper extends existing bounds on flow matching to a deterministic sampling regime, and a time-varying Lipschitz constant of the flow, which allows for analysis of a larger class of distributions (such as those with different support).  The paper largely followed existing flow matching setup (so this was not questions by the reviewers) and focused on novel technical results. The support to the claims was all theoretical (proofs), not numerical. The techniques involved a range of math, from ODE theory (Gronwall's lemma and classical extensions) to probability.  None of the reviewers raised any issues about the correctness.